# Self-Healing Analysis of Half-Warm Asphalt Mixes Containing Electric Arc Furnace (EAF) Slag and Reclaimed Asphalt Pavement (RAP) Using a Novel Thermomechanical Healing Treatment

**DOI:** 10.3390/ma13112502

**Published:** 2020-05-30

**Authors:** José Manuel Lizárraga, Juan Gallego

**Affiliations:** Departamento de Ingeniería del Transporte, Urbanismo y Territorio, Universidad Politécnica de Madrid, 28040 Madrid, Spain; juan.gallego@upm.es

**Keywords:** half-warm mix asphalt, EAF slag, recycled asphalt pavement, microwave heating, mechanical recompaction, healing rates, mechanical properties

## Abstract

Nowadays, the self-healing of asphalt pavements promoted by microwave radiation heating energy is gaining attention and strength in the scientific community. However, most of these studies are only conceptual and, thus, remain shrouded in uncertainty regarding technology development, economy, and application effect. Therefore, there are several efforts underway to offer more effective assisted healing treatments that are capable of overcoming such uncertainties. This paper aims to assess and quantify the healing performance rates (HR) of half-warm recycled asphalt (HWRA) mixtures containing electric arc furnace (EAF) slag and total recycled asphalt pavement (RAP) rates. To this end, a novel assisted thermomechanical healing treatment (i.e., a recompaction-based technique and microwave heating energy) was put forward to promote the potential healing effect of this treatment on the mechanical properties of the asphalt mixtures. In order to do this, three microwave heating temperatures (25 °C, 60 °C, and 80 °C) and three mechanical recompaction levels (0, 25, and 50 gyrations) were selected. After that, the healing performance rates (%, HR) of the asphalt mixtures were calculated by repeated indirect tensile strength (ITS) and indirect tensile stiffness modulus (ITSM). The results indicated that the 8% EAF slag mixture was found to provide significant microwave heating energy savings by up to 69% compared with the benchmark 100% RAP mixture, and, at the same time, it experienced a remarkable stiffness recovery response of 140% of the initial mechanical properties. These findings encourage greater confidence in promoting this innovative thermomechanical-based healing treatment for in-situ surface course asphalt mixtures of road pavements.

## 1. Introduction

Asphalt concrete (AC) mixtures are typically exposed to repeated heavy traffic loading cycles and premature fracture failures caused by thermomechanical surface distress that triggers the appearance of different ways of cracking, causing a significant decrease in the mechanical properties and durability of the asphalt mixtures over time [1,2,3]. In order to overcome these issues, novel assisted healing treatments are gaining attention as an attempt to launch more sustainable engineering solutions that enable them to contribute to extending the service life of existing highway infrastructures that have reached unacceptable driving service conditions and require expensive maintenance intervention costs. Such a healing treatment can, therefore, be seen and conceived as a new step towards a zero-waste concept by enabling it to improve environmental performance, reduce the costs of road pavement construction and maintenance practices, decrease polluting CO_2_ emissions and the exploitation and extraction of natural resources (i.e., virgin aggregates and fresh binder). In general, the use of self-healing pavement techniques has the potential to provide energy consumption savings by 3% (22 GJ) and increase the lifespan of the asphalt pavement by 10%, in comparison with roadways without any self-healing capacity [4,5]. 

Self-healing can be understood as the asphalt concrete’s ability to recover the original mechanical properties as a result of wetting [6], interdiffusion [7,8], and randomization [9,10], which occurs during rest periods [11] due to rearrangement of molecules in the bituminous phase [12]. Indeed, there are two main types of self-healing mechanisms in bituminous mixtures, (i) the adhesive healing at the asphalt–aggregate interface and (ii) the cohesive healing with the asphalt mastic [13]. The cohesive and adhesive bonding interactions of asphalt–aggregate systems are strongly influenced by the Lifshitz–van der Waals interactions, and the acid-base interaction components [14]. Some examples of self-healing asphalt pavement technologies commonly mentioned in the literature review that possess the potential to reverse the fracture failure process and restore the loss in the mechanical performance properties (e.g., stiffness and strength) and functionality suffered by the asphalt pavements during their service life include (but are not limited to these solutions): (1) Encapsulated chemical products containing rejuvenator oil agents to repair broken bonds due to the accumulated micro-crack damage caused by the fatigue cracking process [15,16,17]; (2) magnetic induction heating technique in combination with conductive materials (e.g., steel wool, steel fibers, metal particles, carbon black, and graphite) to speed up the asphalt healing process [18,19,20,21,22,23]; (3) infrared heating technique [24,25,26]; (4) microwave radiation heating energy in combination with ferrous particle aggregates (EAF slag) in order to accelerate the heating and healing capability of the asphalt mixtures [27,28,29,30]; (5) the addition of Nanomaterials into the asphalt mixture design, such as carbon nanotubes (CNTs), carbon black nanoparticles (CBNPs), and graphene nanoplatelets (GNPs); the GNPs nanoparticles especially improve the resistance to permanent deformation, moisture damage, skid resistance [31], flexural strength at low temperatures, and increase the cohesion recovery ratios of the mixtures [32]; and (6) industrial microwave heating applications through a mobile microwave power unit for pavement maintenance works [33,34], with emphasis being placed on recycling techniques for surface layers [35]. 

Nonetheless, there remain some concerns to be addressed regarding durability, life cycle cost, and environmental analysis of the self-healing asphalt pavement techniques mentioned above. For instance, the induction heating technique pioneered by Minsk [36,37], and based on the Joule’s heating effect [38,39], fails to convince researchers, as it causes some adverse effects on the rheological and viscoelastic properties of bitumen due to premature aging; it also produces a detrimental effect on the mixture’s mechanical performance response, as a result of overheating (>110 °C), followed by swelling and drainage of bitumen [40]. Another issue is that the surface of the specimen could be quickly burned because of the skin effect, while the inner temperature of the test specimen might not reach the required target heating temperature [41]. As for the microencapsulation technique, the most significant shortcomings associated with this recent healing pavement technology pertain in part to the impossibility of controlling either the breakage of the microcapsules or the release of the rejuvenator oil agent throughout the compaction process applied by either the steel double-drum or the pneumatic road roller compactor. Whereas, for steel wool fibers, they tend to cluster in balls and increase the air voids content, causing a substantial decrease in the mixture’s mechanical response [42]. As for the GNPs nanoproduct, some authors reported that the nanoparticles tend to present some difficulties in achieving a homogeneous dispersion of GNPs in the asphalt binder as well as some issues related to high production and acquisition costs [32]. 

In this context, the microwave-based radiation energy is becoming the most promising heating and healing treatment due to the provision of an alternating electromagnetic field with a frequency in the order of Megahertz [22,43], which can penetrate throughout the volume of the material to provide rapid and more uniform heating through internal friction of polar molecules [44]. Microwaves are electromagnetic waves with frequencies produce in the range between 100 MHz and 100 GHz [45], while the corresponding wavelengths are from 1 m to 1 mm [46,47]. The electromagnetic wave loss occurs when it passes through materials and causes molecular motion [48]. Indeed, the microwave radiation depends on the strength and the frequency of the electromagnetic field involving dielectric and conductive loss [49], which represent the efficiency of material in absorbing microwave heating energy [50]. In summary, some technical and economic advantages provided by the microwave radiation treatment include: (i) Shorter heating time, (ii) higher heating efficiency without causing overheating of the surface layer of the pavement [34,41,49], (iii) greater economy by enabling to decrease the microwave heating energy consumption within the range of 30–40%, in comparison with the induction heating method [22]; and, finally, (iv) a substantial improvement in the adhesion bonding properties between asphalt binder and aggregates [35], as a result of the work of adhesion described as the necessary energy to break up the aggregate–binder bond at the interface [51]. 

Additionally, the previous advantages provided by the microwave heating treatment can be further enhanced if electric arc furnace (EAF) steel slag aggregates are added into the asphalt mixture design. In fact, steel slag is an industrial by-product material resulting from the steel production process and is widely utilized in the binder and wearing course asphalt mixtures of road pavements, due to its high susceptibility to microwave radiation energy [52,53,54]. Steel slag works as a kind of magnetic material rich in metallic oxides such as MgO, CaO, and Fe_3_O_4_ [55]. In this sense, the replacement of virgin and recycled aggregates with steel slag is gaining momentum and strength as a way to improve some thermal and mechanical properties further (e.g., cohesion strength, stiffness, dynamic modulus, fatigue, and rutting) [56,57,58,59,60], likely due to its excellent roughness, shape, angularity, hardness, morphological, physical, polishing, and wear resistance [53]. Nonetheless, the results reported by other researchers are less conclusive, in the sense that steel slag mixtures exhibited lower mechanical performance (e.g., rutting performance) [58,61] and higher binder consumption [58,62] caused by the aggregate particle size and aggregate-type [63]. Therefore, many of the studies found in the literature review have focused on the heating and healing of mixtures containing only virgin (i.e., aggregates or fresh bitumen) and steel slag aggregates. In this sense, it has been suggested that HWRA technologies with recycled and slag aggregates are promising solutions to increase further environmental, technical, and economic benefits associated with the use of industrial by-products [64,65]. In this regard, González et al. [42] concluded that mixtures with RAP and metal fibers have the potential to increase the self-healing effect on the cracked specimens when using microwave heating energy. However, they found that the addition of RAP material appeared to decrease the healing rates of the asphalt mixtures as they turned out to show a lower susceptibility to microwave radiation energy. Some other researchers investigated the healing effect of adding EAF slag on the mixture’s fatigue cracking resistance. For instance, Bagamapadde and Wahhab [66] and Skaf et al. [53] claimed that the steel slag mixtures were found to offer much lower fatigue life and resilient modulus than that of conventional asphalt mixtures. Nonetheless, Minh Phan et al. [29] evaluated the crack healing performance rate through the three-point bending beam test using microwave heating energy. In particular, they reported that the healing level of mixtures containing 30% of coarse slag was found to be 88% after the second damage-healing cycle. Likewise, Kavussi et al. [67] compared the effect of different mechanical parameters on microwave healing efficiency of asphalt mixtures using semicircular bending beam (SCB) and indirect tensile test (IDT). They found that the recovery of mechanical damage of activated carbon modified asphalt mixes reached a healing rate peak of 63% (SCB) and 58% (IDT) at 25 °C, respectively, and that the healing index had notably decreased as the fracture-healing cycles increased. In this context, the issues related to the provision of microwave healing performance ratios of asphalt mixtures made with RAP and EAF steel slag have led to contradictory and controversial results among researchers, and, therefore, still remain shrouded in uncertainty. 

For all these reasons mentioned above, the primary objective of this research paper is to assess and quantify the potential healing effect of a novel thermomechanical healing treatment (i.e., a recompaction-based technique and microwave radiation energy) on the mechanical performance properties (i.e., repeated indirect tensile strength (ITS) test and indirect tensile stiffness modulus) of half-warm recycled asphalt (HWRA) mixtures containing electric arc furnace (EAF) slag and total recycled asphalt pavement (RAP) rates. 

In this context, the most significant contributions of this research paper to the current state-of-the-art review include the following: (1) The development and startup of a novel assisted healing treatment capable of triggering a recovery/healing effect on the original mechanical properties (i.e., strength and stiffness) of the broken specimens after automated testing; while inducing a similar re-kneading effect on the cracked mixture compared to that used by a vibratory double-drum asphalt road roller compactor in the field; (2) the addition of EAF slag into the asphalt mixture design has been crucial to increasing heating, and healing efficiency, thanks to the use of microwave-based energy treatment; (3) at half-warm temperatures and half of (25) recompaction gyrations applied to the broken specimens, the 4% EAF slag mixture was found to achieve the highest healing performance rate of 140% of internal cohesion strength, whilst, for the reference recycled mixture, this recovery ratio was 93% at 80 °C. As for the stiffness recovery response, the 8% EAF slag mixture exhibited the highest recovery rate of 140% when applying 50 recompaction gyrations, at 80 °C. Therefore, (4) it has been observed that, as microwave heating temperature rises (i.e., from 25 °C to 80 °C), the mixtures with higher steel slag contents reached equivalent, or even much greater, healing ratios than HR = 100%. In contrast, at room microwave heating temperatures (~25 °C), all of the EAF slag mixtures experienced a significant decrease in the healing performance rates, regardless of the number of load cycles applied to the specimens and the content of steel slag added to the asphalt mixture design. Some author’s findings were in line with these findings and agreed with the fact that the optimal microwave heating temperature for a conventional asphalt concrete mixture should be ≤85 °C [68]. As a result of this, one can say that the higher the microwave heating temperature used for the 4% EAF slag mixture, the higher the healing effect, and, thus, the higher the recovery ratios will be on the asphalt mixtures. This temperature-dependent nature was also confirmed by multiple research works that demonstrated that the healing performance rates might begin to increase as heating temperature rises [69,70], causing a significant decrease in the viscosity and flowing of bitumen to heal open cracks more quickly [71]. (5) As for the cohesion recovery ratios set at HR = 100%, it was found that the 8% EAF slag mixture displayed the highest microwave healing efficiency, showing substantial microwave heating energy savings by up to 69% (kWh/kg), in comparison with the benchmark 100% RAP mixture. In contrast, for the stiffness recovery response, the 8% EAF slag exhibited significant microwave heating energy savings by 61% (kWh/kg), in comparison with the recycled mixture. Finally, the bottom-line conclusion is that this novel assisted healing treatment is becoming a revolutionary environmental, technical, and economic solution that can be implemented and upscaled for maintenance and rehabilitation (M&R) works of road pavements. 

## 2. Methodology

### 2.1. Methods

This research study aims to present the healing performance ratios of half-warm recycled asphalt (HWRA) mixtures containing three different electric arc furnace (EAF) steel slag contents (0%, 4%, and 8% EAF) used as partial replacement of recycled RAP aggregates within the particle size of 0/4 mm. The research methodology was broken down into five main phases in order to put across to the readers part of the general idea of what has been done in this research work. Therefore, the first phase consisted of weighing and dosing the corresponding proportions of the mixture components (i.e., steel slag, recycled aggregates, and bituminous emulsion) that were used in the production of the asphalt mixtures, in which all of these mixture components have been mixed and then compacted with the gyratory compactor. Previously, the RAP material was characterized in order to determine the bitumen’s physical consistency properties (i.e., penetration test (dmm) and softening point temperature (°C), and black RAP grading curves. Steel slag aggregates were classified into different particle grain size fractions as follows: 2/4, 0.5/2, 0.063/0.5, and passing through the # 0.063 mm sieve, according to EN 933-1:2012. Part 1: Determination of particle size distribution—Sieving method.

The second phase of this research work consisted in determining the optimum emulsion content (OEC) of the half-warm asphalt mixture with a RAP content equal to 100% using four different emulsion contents (2.0%, 2.5%, 3.0%, and 3.5% over the total weight of RAP) at 0.5% increments. The variation in the emulsion content was to seek for the center of the interval of a dense-graded asphalt concrete mixture within the range of 5 ± 1% air voids, according to Art. 542.5.1.2: Air voids [72]. Therefore, the HWRA 100% RAP mixtures were prepared, according to different emulsion contents previously established, and then compacted with the gyratory compactor device (Cooper Research Technology, Ripley, Derbyshire, UK). That is, the compacted specimens resulted in a diameter of 100 mm and an average specimen height of 60 ± 2.5 mm, applying a mix design compaction energy of 80 gyrations, at 80 °C, and following the standard compaction conditions (α = 0.82°, 600 kPa, and 30 rpm) established by EN 12697-31:2007. Part 31: Gyratory Compactor [73].

Once the cylindrical specimens were compacted, the next step was to calculate the average geometric density resulting from the geometrical dimensions of the bituminous specimens, according to EN 12697-29:2003. Part 29: Determination of the dimensions of a bituminous specimen. To this end, a Starrett^®^ electronic sliding vernier caliper with an accuracy of up to ±0.01 mm, and 0” to 6” (0 mm to 150 mm) measuring range was selected to determine the corresponding specimen thickness, with an accurate tolerance measurement of ±0.1 mm approximately from the edge of the test specimen. The repeatability of the specimen thickness was defined and calculated as the average value of four measures of the test specimens to eliminate the potential differences in terms of width. 

In the third phase, a microwave heating and temperature measurement study was conducted to monitor and plot the surface and internal thermal maps using at least five steel slag aggregate contents (0%, 2%, 4%, 6%, and 8% of EAF) at 2% slag increments. In turn, the mass of the specimens was found to vary from 1000 to 1080 g, likely due to the difference in the apparent gravity between slag and recycled aggregates. The average microwave heating energy consumption (kWh/kg) for each type of slag specimen was calculated and plotted, depending on different microwave heating times within the range of 0 s to 300 s, at 30 s increments. The temperature study was performed using a microwave heating oven with a theoretical output capacity of 700 W of power, and a frequency of 2.45 GHz. This frequency corresponds to an approximate penetration wavelength of 120 mm [74]. However, for this research work, the microwave heating power level was set at 400 W. Once the microwave oven was adjusted to the most convenient heating power level, both internal, and surface heating temperatures (°C) against the microwave heating energy consumed (kWh/kg) for each steel slag content were calculated and plotted. In order to conduct this thermal study, a pocket-sized FLIR C2 thermographic imaging camera (FLIR Systems, Wilsonville, OR, USA) (i.e., adjusted to a temperature emissivity of 0.95, the surface reflexivity temperature of 20 °C, and infrared (IR) resolution of 80 × 60 pixels) was selected in order to draw some conclusions regarding the internal and surface heating temperatures (e.g., mean (µ), standard deviation (SD), and coefficient of variation (CV)) of the test specimens. 

In the fourth phase, the initial mechanical properties of the slag mixtures were determined to evaluate the potential heating effect of adding EAF slag on the mechanical performance response of the asphalt mixtures. In this sense, the EAF slag mixtures were initially tested for stiffness at 20 ºC, followed by repeated indirect tensile strength (ITS) test at 15 °C. The stiffness modulus values were calculated by applying a set of five stress-controlled load pulses at a loading frequency of 2.1 Hz, with a rise time of 124 ± 3 ms, and horizontal maximum peak deformation of 5 µm, according to EN 12697-26:2012. Part 26: Stiffness. Posteriorly, the internal cohesion strength of the asphalt mixtures was calculated by following the indirect tensile strength test at 15 ºC, using a vertical diametral load at a constant deformation rate of 50 ± 2 mm/min, according to EN 12697-23:2008. Part 23: Determination of the indirect tensile strength of bituminous specimens. 

The fifth phase aimed to quantify the healing performance rates (HR) and the healing efficiency of energy (kWh/kg) spent during the microwave heating process. In order to do so, a novel thermomechanical-based healing treatment (i.e., a recompaction-based technique and microwave radiation energy) was put forward to promote the self-healing process and, thus, restore the loss in the mechanical properties of the broken specimens, once they had been subjected to repeated indirect tensile strength (ITS) test and indirect tensile stiffness modulus (ITSM). In order to quantify this recovery, three microwave heating temperatures (25 °C, 60 °C, and 80 °C) and three mechanical recompaction efforts (0, 25, and 50 gyrations) with the gyratory compactor device (α = 0.82°, 600 kPa, and 30 rpm) were selected. Figure 1 presents the main schematic phases of the methodology followed in this research work. The first phase of this research study (a) consisted of material weighing, dosing, and manufacturing, and subsequent compaction of the test specimens; (b) optimization of the mix design via an accelerated curing and drying heating process at 50 °C, for 72 h; (c) the thermographic study plots the surface and internal heating temperature measurements of the rectangular interior section of the test specimens; (d) initial mechanical testing for stiffness and indirect tensile strength test; (e) thermomechanical healing treatment based on a recompaction technique and microwave radiation energy; and (f) 2^nd^-second mechanical testing round applied to the healed specimens and then the calculation of the healing ratios.

### 2.2. Statistical Data Analysis

Once the mechanical response (i.e., strength and stiffness) data were obtained, the next step was to perform a statistical data analysis of grouping results in order to determine the influence of adding EAF slag (independent variable) on the mechanical performance response (dependent variable) of the asphalt mixtures studied. In order to do so, the latest IBM SPSS^®^ (Statistical Psychosis of the Socially Skewed) statistics v26 software (IBM, Armonk, New York, NY, USA) was used and run for such purpose. Consequently, a one-way analysis of variance (ANOVA) [75] was selected for stiffness modulus, whereas, for the ITS values, a rank-based nonparametric analysis through the Kruskal–Wallis H test [76] was chosen. Since the internal cohesion strength values, when graphed, were found not to follow a typical bell-shaped distribution. Indeed, the null and alternative hypotheses were stated in both statistical tests. The null hypothesis is a statement that claims that all samples come from populations with the same medians, while the alternative hypothesis is that not all population medians are equal (i.e., it means that at least one pair of medians is uneven). *p* values < 0.05 were considered significant. Equation (1) shows us how to calculate the Kruskal–Wallis H statistic [76,77]. In turn, there is a correction for ties. The correction factor, C, is given in Equation (2), where the correction is equal to H/C.
(1)$$H=\frac{12}{N\;\left(N+1\right)}\;\;\overset{k}{\underset{i=1}{\sum }}\frac{R^{2}{}_{i}}{n_{i}}-3\;\left(N+1\right)$$
(2)$$C=1-\frac{\sum t_{i}{}^{3}-\;t_{i}}{N^{3}-N}$$
where N is the total number of observations (N = 81), k is the number of groups, n_i_ is the sample size per group (ni = 27), Ri is the sum of ranks for the ith group $\left(\overset{k}{\underset{i=1}{\sum }}\frac{R_{i}{}^{2}}{n_{i}}\right)$, and ti is the count for each tied rank. The degrees of freedom are k (3) − 1 = 2 and α = 0.05 significance level. A Chi-Square distribution approximates the test statistic H. After running some calculations in the Excel^®^ spreadsheet and contrasted in SPSS^®^ after that, the Chi-Square statistic value was found to be 2.8679, and the Kruskal–Wallis H calculated was 7.023. The rejection region for this Chi-Square test is R = {X^2^ : X^2^ > 5.991}. Since 7.023 > 5.991, the null hypothesis is rejected. That is, an H-statistic higher than the rejection region indicates that the null hypothesis is rejected. The *p*-value = 0.03 significance level provides evidence to reject the null hypothesis that the groups have equal ranks, especially the average ranks were as follows: R_i1_ = 48.20; R_i2_ = 43.15; and R_i3_ = 31.65, where the *p*-value means that ITS values do not fit very well; that is, there is not a relationship.

On the other hand, for the ANOVA: single-factor test, a significance level equal to α = 0.05 alongside with a 95% confidence interval were selected as required input data to run this parametric statistical test. Normal distributions, homogeneity of variances, and Welch were ticked in the respective boxes before running the ANOVA test. Post-Hoc comparisons using the Bonferroni correction test and Tukey’s HSD (honestly significant difference) were selected in order to determine which pairs differ significantly. Indeed, there are two statistical parameters (i.e., the *p*-value and the f-ratio) that help us in the decision-making process, either to accept or to reject the null hypothesis as true. The *p*-value identifies the likelihood that a particular outcome may have occurred by chance [75]. In contrast, the F-value represents the ratio between the mean square and the sum of squares and show the influence of the independent variable on the dependent variables (e.g., strength and stiffness) of the asphalt mixtures studied. F-ratio and *p*-value results are shown in Section 6.

### 2.3. Description of the Recompaction Procedure

Due to the fact that there is still no healing and heating technique that is capable of entirely restoring the original mechanical performance properties (i.e., stiffness and strength) of the broken mixtures after automated testing, a new prototype mold was designed and proposed as a means to receive a targeted mechanical re-kneading effect on the fractured specimens. The prototype was launched and put into practice to find out whether this thermomechanical-based healing treatment has the potential or not to reverse the accumulated micro- or macrodamage process suffered by the asphalt specimens—after they have been subjected to a set of repeated load indirect tensile pulses (stiffness), and then followed by a constant deformation tensile strength load of 50.8 mm/min on the specimen’s vertical diametral plane. 

In this context, the recompaction mold is made up of two stretchable and symmetrical faces with a set of six hex head screws (with an outer head diameter of 25.4 mm and inner screw’s thread diameter of 15 mm, with a thread length of 54.05 mm) to ensure complete demolding, assembling, and tightening of the six hex screws of the two symmetrical halves/faces of the mold using, for this purpose, an L-shaped hexagonal Allen wrench of 3/4” Ø (19 mm). This prototype was designed considering a theoretical maximum expandable diameter ranging between 100 and 150 mm, with a total vertical height of about 400 mm, and a clearance between the boundary edge and the container walls of the mold of 40 mm, in which the total weight of the recompaction mold was approximately 30 kg. Figure 2 illustrates the 3D rendering model using SketchUp^®^ Pro v2019 software (Trimble^®^, Sunnyvale, CA, USA). Moreover, a spacer placed at the bottom of the recompaction mold (with a full height of 48 mm and a diameter equal to 100 mm) together with two flat circular metal discs of 5 mm height were selected as part of the study. In particular, one metal disc was placed right below the spacer to fit into the mold, and another one was placed on top of the spacer to level off the surface before and during the recompaction process. After that, a filter paper (with a thickness of 0.1 mm and a diameter of 100 mm) with a thin film of wax was placed above the metal disc to prevent a potential bonding between the spacer and the specimen, while at the same time avoiding possible detachment of slag aggregates from the compacted specimens. As a result, the loss of material might lead to a negative impact on the ultimate geometric density (EN 12697-6:2012. Procedure D: Bulk density by the geometric method) of the test specimens, as it depends on both the thickness and the weight of itself after recompaction. 

Once the mechanical recompaction process was completed, the mold was removed from the gyratory compactor device employing a smooth steel rod (with an arc radius of 18 mm that is tangent to both steel rods with an average thickness of 1/2” (Ø = 12.7 mm) and a total length of 150 mm × 101 mm) integrated and welded to the mold to facilitate its removal from the gyratory compactor device. Posteriorly, a hydraulic piston pushes all of these elements upwards (~400 mm) intending to place the broken specimens on the metal flat discs to give a target re-kneading effect on the fractured samples with the gyratory compactor. Therefore, all of the mold components were previously placed in a forced-draft oven for 1 h to achieve the highest target half-warm temperature of 85 ± 5 °C. This heating mechanism was programmed to take full advantage of the subsequent microwave heating temperatures (e.g., 60 °C and 25 °C) established at the beginning of this study. Nonetheless, in order to keep things warm and, at the same time, avoid a rapid drop in mold temperature (i.e., due to heat diffusion from the metal) as the recompaction process progresses, a portable electric heater of 240 V (with temperature range: 45–95 degrees °F and a thermostat control) was used and adjusted to 25 ± 3 °C, to radiate warmth throughout the laboratory room. Finally, this prototype-mold was designed by the Road Laboratory of the La Politécnica de Madrid and produced in collaboration with Cooper Research technology^®^—Testing Materials in the UK (Appendix A).

### 2.4. Test Procedures

The volumetric characteristics of the mixtures were calculated in terms of maximum density, apparent density, and air voids content. For the maximum density, a couple of pycnometers were selected and set at 4.0 kPa atmospheric pressure, based on the EN 12697-5:2012 standard. Part 5: Determination of the maximum density. Procedure A: Volumetric Method, whereas, the bulk density was calculated by saturated surface dry (SSD) conditions, according to EN 12697-6:2012. Part 6: Determination of bulk density of bituminous specimens. To this end, a set of two non-compacted samples were prepared for each emulsion content, ranging from 2.0% to 3.5%, at 0.5% increments, to determine the maximum density. In contrast, for the apparent density, an average of three cylindrical specimens were prepared (Ø = 100 mm and h = 60 mm height) and then compacted with the gyratory compactor device (EN 12697-31:2007) at 80 °C. Where the difference between the maximum density and the bulk density allows us to determine the content of air voids on the compacted specimens, according to EN 12697:8-2003 standard. Part 8: Determination of void characteristics of bituminous specimens.

Once the volumetric characteristics of the mixtures were calculated, the next step was to determine the mechanical properties of the asphalt mixtures in terms of stiffness modulus, at 20 °C, and indirect tensile strength (ITS) test, at 15 °C. Initially, the load-bearing pavement capacity of the asphalt mixtures was assessed through the stiffness modulus test, at 20 °C, according to EN-12697-26:2012. Part 26: Stiffness. The stiffness modulus was calculated applying a set of five indirect-tensile haversine-shaped load waveform pulses on the vertical diametral section, using a (1) rise time of 124 ± 3 ms; (2) target peak horizontal deformation of 5 µm; (3) loading frequency of 2.1 Hz; (4) peak loading force of 1000 N; and (5) Poisson’s ratio (ν) of 0.35. Thus, 10 load pulses were previously applied to set up the device and system in terms of loading level and frequency. The average stiffness modulus value obtained from the test specimen was repeated and contrasted by rotating it at an angle of 90 ± 10°, around its longitudinal axis on the metal plate of the device. Thus, for an applied dynamic load of P in which the resulting horizontal dynamic deformations are determined, the average stiffness modulus can be calculated using Equation (3):(3)$$S_{m}=\frac{P\left(\gamma +0.27\right)}{t\delta_{h}}$$
where Sm represents the stiffness modulus, MPa; P is the maximum peak dynamic load; N is the γ: Poisson’s ratio; t is the specimen thickness, mm; and $\delta_{h}$ is the total horizontal recoverable deformation, mm.

The second test method used in this research study was the indirect tensile strength (ITSdry), at 15 °C, according to EN 12697-23:2017: Determination of the indirect tensile strength. The internal cohesion strength test consisted of subjecting the cylindrical specimens to compressive strength loads between two loading strips (with a width of 12.7 mm) at a constant deformation speed of 50 ± 2 mm/min. This load provides tensile stress along the vertical diametral plane, causing the failure on this plane. The ITS test was interrupted when the peak compressive strength load dropped by 20% to avoid excessive deformation of the specimens. The indirect tensile strength of the specimens was calculated, in MPa, according to Equation (4):(4)$$ITS=\frac{2P_{max}}{\pi td}$$
where ITS is the horizontal tensile strength expressed in gigapascals or megapascals (GPa, MPa); Pmax represents the ultimate strength load required to fail specimens under diametral compression (kN); t is the specimen thickness, mm; and d is the specimen diameter expressed in terms of mm. Thus, the healing performance rates (HR) were defined by the relationship between the initial and the repeated indirect tensile strength test at 15 °C, as shown in Equation (5):(5)$$\mathrm{HR}\left(\%\right)=\frac{ITS{\;}_{h}}{ITS_{in}}\;\cdot \;100$$
where HR represents the healing or recovery rate shown by the asphalt mixtures after mechanical testing (%). ITSh is the indirect tensile strength of the specimen after being subjected to the thermomechanical healing treatment (MPa). At the same time, ITSin is the initial indirect tensile strength value (MPa).

In this context, a total of 81 cylindrical specimens were prepared and classified into three subset groups. Indeed, a set of 27 specimens were manufactured for each type of steel slag mixture content (0%, 4%, and 8% of EAFS). Then, an average of three specimens was prepared and tested to gain more confidence in the repeatability of the results. Three recompaction levels (0, 25, and 50 gyrations), and three microwave heating temperatures (25 °C, 60 °C, and 80 °C) were chosen as a way to evaluate the susceptibility of the slag specimens to microwave radiation heating energy. Table 1 illustrates the experimental design and test matrix created for this research work, where: Tx °C represents the microwave heating temperature (°C) at which the specimens should be heated and Ni stands for the number of recompaction load cycles applied by the gyratory compactor to the broken specimens. 

## 3. Materials

A cationic slow-setting bituminous emulsion (C67B3) formulated with a penetration residual base binder of 50/70 dmm, with a total residual asphalt binder content of 67% by the total weight of the emulsion, was selected to guarantee a good coating and bonding between recycled and steel slag aggregates with the emulsion. The second material chosen in this research study was the electric arc furnace (EAF) steel slag used as a partial replacement material of recycled aggregates by volume within the particle size fraction of 2/4, 0.5/2, 0.063/0.5, and passing through the # 0.063 mm sieve. What is more, the EAF steel slag had a CE marking as a replacement aggregate for bituminous mixtures and surface treatments for roads, airfields, and other trafficked areas, according to EN 13043: 2002 + AC:2004. The apparent specific gravity of slag was found to fall within the gravity range between 3.51 g/cm^3^ and 3.64 g/cm^3^, as reported elsewhere [78]. The particle size distribution and the grading curve of steel slag resulting from the screening and sieving of the fine fraction of this residue are shown in Table 2 and Figure 3. In contrast, the chemical composition of steel slag is collated in Table 3. 

On the other hand, the recycled asphalt pavement (RAP) was homogenized, quartered, treated, and classified into two convenient size fractions; that is, a fine fraction (0/5 mm) and a coarse fraction (5/25 mm). The RAP proportion was determined to be 60% in the fine fraction (0/5 mm) and 40% in the coarse fraction (5/25 mm). Figure 4 depicts the black RAP grading curves for both recycled RAP fractions; that is, the blue dash line represents the coarse fraction, while the orange sold line belongs to the fine fraction of this material. Following the RAP proportion, the particle grain size distribution of the RAP material falls within the RE2 threshold values established in Art. 20 of PG-4 “Cold in-place recycling of bituminous mixtures with bitumen emulsion” [79]. 

The recycled binder was extracted from the RAP material to be characterized in terms of residual binder content using a rotary evaporator by the centrifuge extractor method (EN 12697-3:2013. Part 3: Bitumen recovery. Rotary evaporator). Moreover, the bitumen’s physical consistency properties were determined through the penetration test (dmm) and the softening point temperature (°C), as collated in Table 4.

## 4. Mix Design

### 4.1. Determining Optimum Emulsion Content

In order to determine the optimum emulsion content (OEC) of the preliminary mixture design, the mixtures were manufactured by heating the emulsion at 65 °C, recycled and steel slag aggregates at 100 °C, blending basket at 95 °C, and the cylindrical mold (Ø = 100 mm) at 100 °C, for 2 h. All of these materials selected (e.g., EAF slag, RAP aggregates, and cationic emulsion) were subsequently poured in the blending basket and then mixed for 3 min, at 80 rpm, aiming to ensure the proper mixing, coating, and bonding between recycled and slag aggregates with the cationic emulsion. The next step consisted of preparing a set of three asphalt specimens (with a diameter of 100 mm and 60 ± 2.5 mm height) for each emulsion content (2.0%, 2.5%, 3.0%, and 3.5% o/RAP aggregates) using a mix design compaction energy of 80 gyrations, at 80 °C. The compaction process was conducted following the standard compaction conditions (α = 0.82°, 600 kPa, and 30 rpm) established by EN 12697-31:2007. Part 31: Specimen preparation by the gyratory compactor. The compaction curves of the test specimens were obtained from an average of three samples and then monitored through the change of the specimen thickness, according to the EN 12697-10:2003. Part 10: Compactibility. Posteriorly, a set of 12 cylindrical specimens was placed into a forced-draft convection oven for three days (72 h), at 50 °C, in order to be subjected them to an accelerated curing and drying treatment to reproduce as far the mixing and field compaction conditions as possible, according to Spanish technical specifications [79]. Therefore, as soon as the samples reached constant weight in the forced-draft oven, the specimens were taken out of the oven and kept at room temperatures (25 °C), for 48 h, in the laboratory.

Once the asphalt specimens were compacted and subjected to a curing process, the next step was to determine the volumetric characteristics of the asphalt mixtures in terms of geometric and apparent density, by saturated surface dry (SSD) conditions, and air voids content (EN 12697-8:2003. Part 8. Determination of void characteristics of bituminous specimens). The former apparent density by SSD was considered as benchmark density to calculate the number of air voids on the compacted specimens. Table 5 shows the volumetric characteristics of the preliminary asphalt mixture design using four emulsion contents, ranging from 2.0% to 3.5%, at 0.5% increments. In this regard, the optimum emulsion content (OEC) of the asphalt mixture design was defined, depending on the target air voids criterion (Vm = 5.0%), i.e., aiming at the center of a dense-graded asphalt concrete mixture, according to Spanish technical regulations [72]. Thus, 2.6% (o/RAP aggregates) of asphalt emulsion was found to meet the target air voids criterion, i.e., this emulsion content was found to fall right within the center of the range between 4.0% and 6.0% of air voids. Figure 5 shows the linear relationship between asphalt emulsion content and air voids content. That is, the y-Axis shows the air voids content calculated by linear interpolation with a fitted regression coefficient (R^2^) of 0.99. At the same time, the x-Axis displays the content of asphalt emulsion added to the preliminary mixture design. As a result of the work done, 2.6% (o/RAP aggregates) of asphalt emulsion made up of 67% of residual bitumen content has been selected. So, 1.742% (2.6% o/RAP * 0.67%) of residual base binder wrapped by the cationic emulsion was added to the 100% RAP mixture design. The resulting asphalt binder content of the 100% RAP mixture was the sum of 1.742% + 4.89% = 6.63% over the total weight of RAP aggregates.

### 4.2. Mixture Composition

Once the bitumen emulsion content was partially calculated, the next step was to recalculate the target effective emulsion content that should be added to the asphalt mixture design. This recalculation process had to be conducted because the fine fraction of steel slag tends to absorb a slightly higher effective bitumen emulsion content (i.e., due to a higher porosity structure and a greater specific surface area of this residue) than that of the recycled aggregates since they are already covered and coated by a thin film of aged asphalt binder. Steel slag was selected as a partial replacement material of recycled aggregates in the particle size fraction (2/4 mm, 0.5/2 mm, 0.063/0.05 mm, and passing through the # 0.063 mm sieve) of the mixture design. Though some researchers claim that the replacement of the coarse slag fraction improves the mechanical properties of the asphalt mixtures [53], in this research, only the fine fraction of slag was chosen since this particle aggregate size offers rapid and more homogeneous heating throughout the volume of the specimen [80]. In this sense, the absorption coefficient of the recycled aggregates was 0.75, whereas, for the EAF slag, this coefficient was found to be 3.0. Equation (6) shows how to recalculate the optimum bitumen emulsion (OBC) content needed to counteract the absorption of slag in the asphalt mixture design: (6)$$OBC_{\%slag}=\%\;OBC_{100\%RAP}+\left(3-0.75\right)\cdot \;\frac{\%slag}{100}$$
where OBC_%slag_ (% o/RAP) is the optimum binder content for a mixture with a certain percentage of slag. OBC100%RAP (%, o/RAP) represents the optimum asphalt binder content for a mixture with 100% RAP. The term (3 − 0.75) × %Slag/100 in Equation (6) stands for the increase in the asphalt binder content needed to counteract the higher absorption of slag in the asphalt mixture design. Therefore, the increase in asphalt emulsion content of the mixture design was calculated as follows: 2.6% + (3 − 0.75) × % Slag/100. For example, if the 100% RAP mixture is replaced with 6%/slag of the total volume of the mix design, the additional emulsion content needed to counteract the absorption of slag in the mixture design containing 6% EAF and 94% RAP would be expressed as follows: OBC6%slag = [2.6% + (3 − 0.75) × 6%slag/100)] = 2.6% + 0.135% = 2.735%o/RAP. 

In this context, the final compositions of the steel slag mixtures (0%, 2%, 4%, 6%, and 8% of EAFS), together with the corresponding emulsion content added to the preliminary mixture design, are collated in Table 6. Since the specific gravity of steel slag (3.64 g/cm^3^) is quite different from the siliceous recycled RAP aggregates (2.74 g/cm^3^) [61], the equivalent-volume replacement principle was selected in order to substitute the recycled aggregates in the fine fraction that passes through the 4.0 mm sieve, in which the replacement was made by volume to maintain the same proportions in the particle mineral skeleton. Afterward, the specimens were manufactured and compacted, applying 80 gyrations to 920 g of RAP and 120 of EAF slag to achieve a total height of approximately 58.5 ± 1.5 mm. It was observed that the compaction curves of the slag mixtures showed a slightly higher level of densification than that of the reference recycled mixture during the first phase of compaction, thus having much greater workability than the other EAF slag mixtures. In addition, the most convenient compositions added to the mixture design include: (1) For the 8% EAF slag, 5% of slag was added (i.e., passing # 4 mm and retained in #2 mm sieve) and 3% passing through the # 2 mm sieve; (2) for the 6% EAF slag, 4% of slag was used in the 2/4 mm sieve, and 2% for the 0/2 mm sieve; (3) for the 4% EAF slag, 2.5% of slag was dosed for the 2/4 mm sieve, and 1.5% EAF slag for the 0/2 mm sieve; and (4) for the 2% EAF slag, 1.5% of slag was chosen for the 4/2 mm and 0.5% for the 0/2 mm sieve. 

## 5. Thermographic Study

In order to determine the optimum microwave heating temperature/time/cost associated with each steel slag mixture, an average of three cylindrical specimens was prepared (with a diameter of 100 mm and a total height of 60 ± 2.5 mm) for each steel slag mixture content (0%, 4%, and 8% of EAFS) as a way to minimize the potential variations in terms of test results. Hence, the semicylindrical test specimens were cut into two semicylindrical halves, with an electric saw cutting machine, on the vertical diametral plane in order to facilitate the inner temperature measurement of the specimens. To this end, a microwave heating oven with a theoretical output capacity of 700 W of microwave power (productive) was selected for this temperature study. However, the microwave power was adjusted to produce 400 W, with a frequency of 2.45 GHz, where the optimum heating time is highly dependent on the input power and energy efficiency [41]. Therefore, the test specimens were microwave heated for different fixed heating times in the range of 0–300 s, at 30 s increments. Previously, a piece of rectangular-shaped cardboard (with a total length of 200 mm, 160 mm wide, and a total thickness of 3.5 mm) was placed just right below the semicylindrical test specimen to prevent the transfer of heat energy by conduction from the glass tray, attached to turntable support, towards the test specimen. Following the microwave heating process, both surface and internal heating temperatures were monitored and controlled using a pocket-sized FLIR C2 thermal imaging camera to obtain the most representative thermal statistical heating data (e.g., mean (µ), standard deviation (SD), and coefficient of variation (CV)) derived from the inner rectangular section of the test specimens. Both surface and internal heating temperatures of the test specimens were measured every 30 s of heating, at which four heating temperature measurements were taken randomly from the inner and surface section of the test specimens so that the most uniform temperature can be obtained and plotted after that. 

In this context, the temperature study revealed that the specimens showed a slightly higher internal heating temperature than that of the surface temperature, likely caused by a higher heat-flow diffusion in the periphery of the samples that enabled them to achieve the target internal heating temperatures more quickly [81]. On the other hand, it is worth noting that if the heating of RAP binder got to be too fast and exceeded the melting point temperature, the mechanical performance properties (i.e., strength and stiffness) of the asphalt mixture may be affected and, hence, began to decrease its mechanical response caused by the hot gases released and trapped inside the cylindrical specimen that cannot be dissipated quickly enough. These issues may lead to swelling, tenderness, and crumbling of the test specimens, causing a detrimental effect on the mechanical response and, hence, a decrease in the self-healing rates of the asphalt mixtures. Figure 6 illustrates how the test specimen is initially cut into two halves and then placed into the microwave cavity to conduct the microwave temperature study: picture (a) shows the placement of the test specimen in the microwave oven set at 400 W of power; thermography (b) shows the specific specimen surface temperature using a pocket-sized FLIR C2 thermal imaging camera, and thermography (c) illustrates how the two halves of the test specimen are opened and separated to measure the internal heating temperature of the rectangular central section of the test sample. 

Figure 7 illustrates the microwave heating energy consumption (kWh) resulting from the surface of the test specimens. Indeed, the Y-axis exhibits internal heating temperature (°C) of the rectangular central section of the test specimens, whilst the x-Axis plots the microwave heating energy consumed (kWh/kg) from the mixtures made with five slag contents (0%, 2%, 4%, 6%, and 8% of EAFS). Linear regression models (0.99 ≥ R^2^ ≥ 0.95) and R-squared were used and depicted to predict the relationship between internal and surface heating temperature (°C) with the microwave heating energy consumed (kWh/kg) from the asphalt mixtures during microwave heating.

As a result of the microwave heating and temperature measurement analysis, the optimal microwave heating times resulting from the internal rectangular section of the 0% EAF slag (100% RAP) specimens were as follows: 220 s (80 °C), 130 s (60 °C), and 15 s (25 °C). In contrast, for the 4% EAF slag, times required to reach these internal temperatures were: 160 s (80 °C), 90 s (60 °C), and 12 s (25 °C); while, for the 8% EAF slag, the microwave heating times were: 120 s (80 °C), 70 s (60 °C), and 12 s (25 °C). Throughout the microwave heating process, the heating energy consumed from the microwave was monitored and controlled using an Efergy^®^ electricity monitoring device plugged to the electric current supply of 250 V; where the average microwave heating energy consumption reflected on the display of the electricity meter was found to be 0.0075 kWh/kg. This ratio has been measured every 30 s of heating, setting the microwave oven to 400 W of power. In this line, one can say that the heating radiation energy spent by the oven was found to be more dependent on the input power of the microwave than on the slag content added to the asphalt mixture design. However, the temperature of the specimen did prove to be highly dependent on the steel slag content added in the mixture design. 

As can be observed in Figure 8, the average microwave heating energy consumption of the 8% EAF slag of 0.028 kWh/kg was obtained at 80 °C, whereas, for the 0% EAF slag, this ratio was 0.051 kWh/kg. Therefore, to achieve the same heating temperature of 80 °C, the mixture with 8% EAF slag experienced a substantial decrease in the microwave heating energy consumption of −0.023 kWh/kg (ΔEs = −45.1%), in comparison with the reference recycled mixture. This rapid and uniform heating could be attributed to the higher susceptibility of steel slag to microwave radiation energy, and the proper distribution of slag residue in the entire mass of the specimen throughout the mixing process in the bowl (EN 12697-35:2004 + A1:2007. Part 35: Laboratory mixing). On the other hand, by comparing the surface with the internal heating temperature graph, one can say that the surface of the specimens tends to spend a slightly higher microwave heating energy than that of the inner section. In other words, the average internal heating temperature turned out to be about 11 °C (SD = 1.8 °C) higher than that of the surface temperature due to the faster heat dissipation at the specimen’s surface. Other researchers reported a similar heating trend regarding the variation in the thermal gradient obtained from the internal and the surface heating temperatures of the test specimens [41]. Therefore, as a general trend in the heating graphs, it can be said that the higher the inclusion of steel slag aggregates in the asphalt mixture design, the faster the target microwave heating temperature can be reached, and, hence, the more the microwave heating energy can be saved for in-situ pavement maintenance works. 

## 6. Test Results

### 6.1. Stiffness Modulus of the Asphalt Mixtures

Figure 9 illustrates the repeated indirect tensile stiffness modulus (ITSM) values resulting from the half-warm recycled asphalt (HWRA) mixtures made with steel slag and recycled aggregates. Indeed, the average stiffness modulus values of the 100% RAP mixture of 6343 MPa was obtained (with a standard deviation (SD) of 533 MPa, and a coefficient of variation (CV) of 8.41%), whereas, for the 4% EAF slag, the load-bearing pavement capacity was 6081 MPa (SD = 631 MPa, and CV = 10.38%) and, for the 8% EAF slag, this value was 5564 MPa (SD = 703 MPa, and CV = 12.63%). Therefore, a significant decrease in the stiffness modulus values of the 4% EAF slag of 262 MPa (Δsm = −4.13%) was obtained, while for the 8% EAF slag, this reduction was found to be 779 MPa (Δsm = −12.28%), in comparison with the reference recycled mixture. This decreasing effect on the mixture’s mechanical response was consistent with the results released by other researchers [62]. However, the decrease in the average stiffness modulus values of the 4% and 8% EAF slag mixtures can be assumed as a positive aspect to improve the mixture’s fatigue cracking resistance, since it would make it less stiff by enabling greater deformations before its fatigue cracking failure occurs in the field [64]. 

In order to contrast the trueness of the results, a one-way analysis of variance (ANOVA: Single-factor) was performed using an Excel^®^ spreadsheet and contrasted with SPSS^®^ software after that. Normality and homogeneity of variance were reviewed and then ticked in the respective boxes. A significance level of α = 0.05 and a 95% confidence interval were also selected as input data required to run such a parametric statistical test. The analysis showed that the effect of adding EAF slag on the mechanical response of the mixtures was significant; that is, Fratio was found to be F(2,78) = 9.105, and *p*-value = 0.00279 (*p*-value < 0.05). The rejection region for this F-test was R = {F: F > Fc = 3.114}. The F-ratio was 9.105 > Fcritic = 3.114. Thus, Fratio showed that there was enough evidence to claim that not all three-population means were equal. As a result of this, one can conclude that the null hypothesis is rejected, based on the *p*-value and Fratio calculated. In other words, the addition of EAF steel slag in the particle size fraction of steel slag (passing through the # 4 mm sieve) had a significant impact on the mechanical performance response (stiffness) of the asphalt mixtures studied. 

### 6.2. Influence of Slag on the Healing Ratios

The healing performance rates arising from the repeated indirect tensile stiffness modulus are shown in Figure 10. Indeed, the x-Axis (horizontal) depicts the microwave heating energy consumed (kWh/kg), depending on the target microwave temperature established (25 °C, 60 °C, and 80 °C), for each slag mixture content (0%, 4%, and 8% of EAF). In contrast, the y-Axis exhibits the healing performance rates (HR) plotted on the vertical axis as a percentage of recovery (HR = %). The HR values represent the average of three asphalt specimens, and the error limit bars correspond to the standard deviation (SD) of the data samples from their sample mean. Where the microwave healing efficiency of the asphalt mixtures was determined according to the measurement unit (°C/kWh/kg). In this line, two different criteria were adopted as a way to compare the healing efficiency provided by the slag mixtures, i.e., a target microwave heating temperature (60 °C and 80 °C) and a required percentage of healing rate (HR = 75% and HR = 100%).

As is illustrated by Figure 10a, the slope of the healing rates of the 0% and 4% EAF slag were found to fall below 30% regardless of the microwave heating temperature selected. These results were in line with the recovery ratios reported by other researchers. For instance, Wang et al. [41] indicated that the self-healing rates of asphalt concrete (AC) mixtures were found to fall within the healing rate values between 12% and 18% when utilizing only one microwave damage-heating and healing cycle—and that the error bar limits had followed similar trends. Nevertheless, the slope of the healing curve of the 8% EAF slag mixture showed a steep upward behavior by reaching a maximum healing peak rate of 52% at 80 °C (SD = 8.87%) and an average microwave heating energy consumption of 0.0381 kWh/kg. Other researchers also confirmed this upward healing trend. They reported that the temperature sensitivity of the healing rates was nonlinear, and the recovery ratios had increased due to the increase in temperature, especially for mixtures with higher bitumen content [69]. As a result, one can say that the use of one microwave damage-heating and healing cycle, at 80 °C, was found to be sufficient to restore half of the original mechanical properties lost (HR = 51.7%) by the broken asphalt specimens, in comparison with the reference recycled mixture. 

From the 25 recompaction gyrations graph, it was observed that the 4% EAF slag mixture showed the highest healing rate of 125.4% (SD = 6.5%, CV = 13.23% and Ec = 0.04711 kWh/kg). In contrast, for the 8% EAF slag, this recovery ratio was 121.2% (SD = 0.065%, CV = 5.66% and Ec = 0.0381 kWh/kg). Therefore, a considerable decrease in the average microwave heating energy consumption of the 8% EAF slag of 19.13% (ΔEAF8/4% = −0.00901 kWh/kg) was obtained, in comparison with the 4% EAF slag, as seen in Figure 10b. Meanwhile, for the 50 recompaction gyrations graph, the 8% EAF slag exhibited the highest healing rate of 139.2% (SD = 0.082, and CV = 6.24%) at 80 °C and an average microwave heating energy consumption of 0.0381 kWh/kg. In contrast, for the 4% EAF slag, this recovery ratio was found to be 138% (SD = 2.92%, and Ec = 0.0471 kWh/kg), whereas, for the benchmark recycled mixture, this healing percentage was found to be 111% (SD = 14.1%, and Ec = 0.0642 kWh/kg), as illustrated in Figure 10c. As a general trend shown in Figure 10c, one can say that the higher the number of mechanical recompaction gyrations applied to the optimal microwave heating temperature, the more the healing efficiency and the recovery ratios of the asphalt mixtures.

Additionally, the increase in the average healing performance rates could also be attributed to an increase in the average geometric density (g/cm^3^) values of the asphalt mixtures. In other words, the average geometric density value of the 8% EAF slag mixture was found to increase from 2.230 g/cm^3^ to 2.332 g/cm^3^ (Δ$\rho b_{dim}$= 0.102 g/cm^3^) when applying 50 recompaction gyrations, at 80 °C. Therefore, a significant reduction in the number of air voids of the 8% EAF slag of 4.3% was obtained, causing a considerable decrease in the amount of air voids content from 7.8% to 3.5%. The change in the specimen thickness varied from 58.65 mm to 55.46 mm. The air voids content of the test specimens was calculated based on the average maximum density value of 2.417 g/cm^3^. In this line, some researchers emphasized that the air voids content plays a crucial role in the healing effect on the damaged asphalt specimen by improving the internal pressure and mobility of bitumen, causing the asphalt binder to reduce its viscosity and flow through the cracks easier [22]. In summary, the increase in the healing performance rates was influenced by the mixture’s volumetric characteristics (i.e., air voids and apparent density) as well as the amount of higher residual bitumen content added to the mixture design. Therefore, the 8% EAF slag mixture was found to deliver the highest possible recovery ratios (i.e., 2.79% × 0.67 = 1.8693% + 4.89 (%/o/RAP) = 6.76%) because of the higher bitumen content, in comparison with the remaining steel slag mixtures studied.

### 6.3. Indirect Tensile Strength of the Asphalt Mixtures

In Figure 11**,** the repeated indirect tensile strength (ITSf) value results arising from half-warm mixtures made with three EAF slag contents (0%, 4%, and 8% EAF) were calculated and plotted. The internal cohesion strength values of the slag mixtures were determined, according to EN 12697-23:2017: Determination of the indirect tensile strength of bituminous specimens. In particular, the average in-dry indirect tensile strength values of the HWRA 100% RAP (0% EAFS) mixture was 2.186 MPa (SD = 0.352 MPa, CV = 16.14%, Variance (Var) = 0.1106), whereas, for the 4% EAF slag, the average internal cohesion value was 2.056 MPa (SD of 0.152 MPa, CV = 7.41%, Var = 0.021) and for the 8% EAF slag, this number was 1.947 MPa (SD = 0.161 MPa, CV of 8.24% and Var = 0.0229). Therefore, a slight decrease in the cohesion strength values of the 4% EAF slag of 0.130 MPa (ΔITS = −5.95%) was obtained, whereas for the 8% EAF slag, this number was 0.239 MPa (ΔITS = −10.92%), in comparison with the reference recycled mixture. 

In order to contrast these results, a rank-based nonparametric test using the Kruskal–Wallis H test was conducted to evaluate further the potential impact of adding EAF steel slag on the mechanical response and healing rates of the asphalt mixtures. To this end, a significance level of α = 0.05 and a 95% confidence interval were selected to perform such a statistical test. In this case, after running some calculations, the *p*-value was found to be 0.02985, and the asymptotic significance level was found to be 0.011. The *p*-value was less than the significance level (*p*-value < 0.05), so the null hypothesis was rejected based on the *p*-value calculated. Therefore, one can say that there is enough evidence to claim that some of the population medians are unequal at the alpha level of α = 0.05. In a nutshell, the analysis showed that the replacement of steel slag by volume in the asphalt mixture design has a significant impact on the internal cohesion strength values of the asphalt mixtures.

In this context, one can say that the higher the content of steel slag in the asphalt mixture design, the lower the internal cohesion values of the asphalt mixtures. The likely explanation for this downward trend is due to decreased interaction and chemical affinity (i.e., the ratio offered between CaO and SiO_2_ content) between recycled and slag aggregates with the residual bitumen emulsion [61], causing a detrimental effect on the mixture’s mechanical strength resistance [48,82]. Despite this, one can say that the average internal cohesion values shown by the steel slag and recycled mixtures were found to meet the minimum in-dry indirect tensile strength (ITSdry > 1.7 MPa) values stipulated by the Spanish technical specifications, according to Art. 20 of PG-4 “Cold in-place recycling of bituminous mixtures with bitumen emulsion” [79]. 

### 6.4. Influence of Slag on the Healing Ratios

Figure 12 illustrates the cohesion healing performance rates (HR) and the microwave heating energy consumed (kWh/kg) from the repeated indirect tensile strength test. The solid blue healing curve represents the 0% EAF slag; the orange healing curve shows the recovery ratios belonging to 4% EAF slag, while the red curve presents the recovery rates for the 8% EAF slag. Indeed, these healing rate curves are a parameter of the healing efficiency of the energy consumed from the microwave heating process [32]. In particular, in Figure 12a, the slope of the healing curve of the 100% RAP mixture made with siliceous recycled aggregates remained at the same level of healing, regardless of the microwave heating temperature selected. In contrast, the healing curves of the 8% EAF slag rocketed to a self-healing rate peak of 56.3% (SD = 3.45, and Ec = 0.02022 kWh/kg), then it levelled off at 80 °C (HR = 56.2%, SD = 5.91%, and Ec = 0.0381 kWh/kg). Therefore, a significant average microwave heating energy consumption of 8% EAF slag of 46.93% (ΔEc = 0.01788 kWh) was obtained, in comparison with the 100% RAP mixture. The main reason that the healing level remained stable is likely due to the chemical damage suffered by the asphalt bitumen because of overheating [83,84]. This stabilization trend could also be attributed to the theory that exists a critical temperature (related to near-Newtonian behavior temperature of asphalt binder) above which healing levels and optimum microwave heating temperature decreases the healing effect on the mixture’s mechanical response [41]. 

As is illustrated by the 25 recompaction gyrations graph, the slope of the 8% EAF slag mixture followed a steep upward behavior by soaring up from 52.29% at 25 °C (SD = 3.98%) to 100.2% at 60 °C (SD = 13.17%) and then it leveled off at 80 °C (HR = 108% and SD = 20.32%). Figure 12b,c depicts the rate at which the 4% EAF slag mixture experienced a similar upward healing trend, rising sharply from 53.5% at 25 °C to 139.9% at 80 °C. In this sense, it can be seen from this graph that only the mixtures with higher steel slag contents (4% and 8% EAFS) reached equivalent healing levels, or even greater, to HR = 100%. As a result, the 4% EAF slag was found to spend an average microwave heating energy consumption of 0.0261 kWh/kg, whereas, for the 8% EAF slag, this energy ratio was 0.020 kWh/kg. In terms of energy savings, a remarkable increase in the average microwave heating energy consumption of 4% EAF slag of 30% (ΔEc = +0.0061 kWh/kg) was obtained. This increase in energy represents the amount of microwave heating energy needed to achieve and trigger the same healing effect (HR = 100%) on the 4% EAF slag mixture. However, at 80 °C, the 4% EAF slag mixture exhibited the highest healing performance rate of 139.88% (SD = 17.57%), with an average microwave energy consumption of 0.04711 kWh/kg. 

In this context, as can be observed in the 50 recompaction gyrations graph, the slope of the healing curve of the 4% EAF slag mixture was steeper, soaring sharply at 60 °C, and kept growing after that. Meanwhile, for the 8% EAF slag mixture, the slope of the healing curve reached a peak of 122.51% (SD = 16.96%) and an average microwave heating energy consumption of 0.03813 kWh/kg. In contrast, the 4% EAF slag exhibited a similar trend in terms of healing rate; that is, this mixture displayed a steady rising behavior by hitting its highest point of healing of 135.5% (SD = 3.8%), and an average microwave heating energy consumption of 0.04711 kWh/kg, as illustrated in Figure 12c. 

As a result of the healing levels obtained, at 80 °C, the average geometric density value of the 0% EAF slag mixture was found to be 2.415 g/cm^3^ (with an SD of 22.85 g/cm^3^ and a CV equal to 0.95%), and an average specimen thickness of 54.05 mm after applying 50 recompaction gyrations. In contrast, for the 4% EAF slag, the average geometric density value was 2.369 g/cm^3^ (SD = 43.84 g/cm^3^, CV = 1.85%, and h = 56.44 mm), while, for the 8% EAF slag, the specimen densification value was 2.332 g/cm^3^ (SD = 15.1 g/cm^3^, CV = 0.65% and h = 57.82 mm). Although the 100% RAP mixture reached its highest point of densification ($\gamma_{b,\;dim}\;$= 2.415 g/cm^3^), the slope of the healing curve grew dramatically from 50% at 25 °C (SD = 10%) to 112.9% at 80 °C (SD = 6.26%), using an average microwave heating energy consumption in the order of 0.004 kWh/kg and 0.0642 kWh/kg. As for the cohesion recovery response values, one can say that the higher the average geometric density of the mixtures, the lower the content of air voids reached during the mechanical recompaction process, and, therefore, the greater the healing effect and the recovery ratios of the mixtures.

Table 7 illustrates the healing efficiency (kWh/kg) associated with the target healing rates (e.g., HR = 100% and HR = 75%) for each EAF slag mixture content (0%, 4%, and 8%). For example, in order to reach a cohesion recovery rate of HR = 100%, the mixture with 8% EAF slag and 92% RAP showed the highest microwave heating energy saving of 69% (Ec = 0.020 kWh/kg), in comparison with the 100% RAP mixture (Ec = 0.0642 kWh/kg). On the other hand, the electricity unit price per kWh offered in the energy market supply by Endesa^®^ power company is roughly 0.1198 kWh [85]. Therefore, for the 8% EAF slag mixture, a significant decrease in the cost of microwave heating energy (ΔEs = 0.0442 kWh/kg × 0.1198 kWh) of 0.005295 (kWh/kg) was obtained, in comparison with the benchmark 100% RAP mixture. In other words, if the 8% EAFS slag mixture is used, approximately more than half of the cost of microwave heating energy will be saved for pavement maintenance works. 

## 7. Conclusions

A novel assisted healing treatment was launched and put into practice to quantify the healing performance rates arising from the mechanical performance properties of half-warm recycled asphalt (HWRA) mixtures containing three electric arc furnace (EAF) steel slag contents (0%, 4%, and 8% of EAFS) and total recycled asphalt pavement (RAP) rates. This innovative healing treatment was not only used to evaluate the heating and healing efficiency caused by the addition of higher EAF slag contents during microwave heating, but rather to demonstrate the tremendous potential healing effect of this treatment on the mechanical performance response (i.e., indirect tensile strength and stiffness modulus) and the recovery ratios of the asphalt mixtures. In this regard, we have determined which slag mixture content has been more likely to recover the original mechanical properties after mechanical testing and with the lowest possible microwave heating energy consumption. Thus, as a result of the laboratory work done, the most significant conclusions and findings that can be drawn from this research paper are listed below:The addition of 8% EAF slag into the asphalt mixture design was found to be the most energy-efficient solution by enabling it to speed up the increase of the specimen temperature and, at the same time, optimize the healing efficiency during microwave heating.Though the EAF slag mixtures exhibited a slightly lower mechanical response than that of the reference 100% RAP mixture, the 4% and 8% EAF slag mixtures met the minimum indirect tensile strength (ITSdry ≥ 1.7 MPa) values stipulated by the Spanish technical specifications. What is more, the EAF slag mixtures showed similar stiffness modulus values compared to those exhibited by conventional asphalt concrete (AC) mixtures made with a 50/70 pen grade bitumen.It was observed that the reference mixture made with siliceous recycled aggregates exhibited much lower susceptibility to microwave radiation heating energy since they reached recovery ratios below 30%—when only one damage-healing cycle is applied. Nevertheless, the 8% EAF slag mixture exhibited a substantial improvement of 52% in the healing performance rates when the temperature increases from 25 °C to 80 °C.At 80 °C, the 4% EAF slag mixture showed the highest ITS recovery ratios of 140% when half of the recompaction energy is applied. In other words, 25 recompaction gyrations were found to be sufficient to produce a similar, or even higher, healing effect on the broken specimens in comparison with fifty (50) recompaction gyrations.It was found that the optimal inclusion of EAF slag content into the asphalt mixture design, together with the microwave heating temperature, played an even more critical role in the provision of the healing rates than the number of recompaction gyrations applied to the broken specimens. What is more, if the microwave heating temperature of the 8% EAF slag mixture increases above 60 °C, the recovery ratios will remain at the same level of healing performance. Nonetheless, the recovery of the internal cohesion of the 4% EAF slag mixture was found to show a linear and steep upward behavior as the microwave heating temperature increases. Therefore, it would be interesting to determine the optimal heating temperature at which the healing curve reaches its maximum peak, stabilize, and fall again.As for the cohesion recovery ratios set at HR = 100%, the mixtures made with 8% EAF slag and 92% RAP exhibited the highest healing efficiency by showing substantial microwave heating energy savings by 69% for ITS, whereas, for the stiffness recovery response, the energy-saving ratio was 61% of kWh/kg, in comparison with the reference recycled mixture.

The results obtained in this research work are promising in demonstrating the technical and economic feasibility of using a novel thermomechanical-based healing treatment for mixtures requiring lower manufacturing, mixing, and compacting temperatures in the laboratory. In particular, fatigue cracking resistance and moisture susceptibility of the slag mixtures should be further investigated in order to gain more confidence in using this assisted healing treatment for other mechanical tests. 

Finally, the authors hope that this research work serves as a starting point to develop portable microwave heating equipment as a new step towards promoting this heating technology for its use in small- or full-scale maintenance applications, such as repairing deteriorated urban pavement sections or federal highways that have reached unacceptable driving service conditions. Therefore, the construction and monitoring of urban test sections made from industrial by-products (i.e., recycled and slag aggregates) are highly recommended to evaluate the potential development and transfer of this technology from the lab to full-scale works.

## Figures and Tables

**Figure 1 materials-13-02502-f001:**
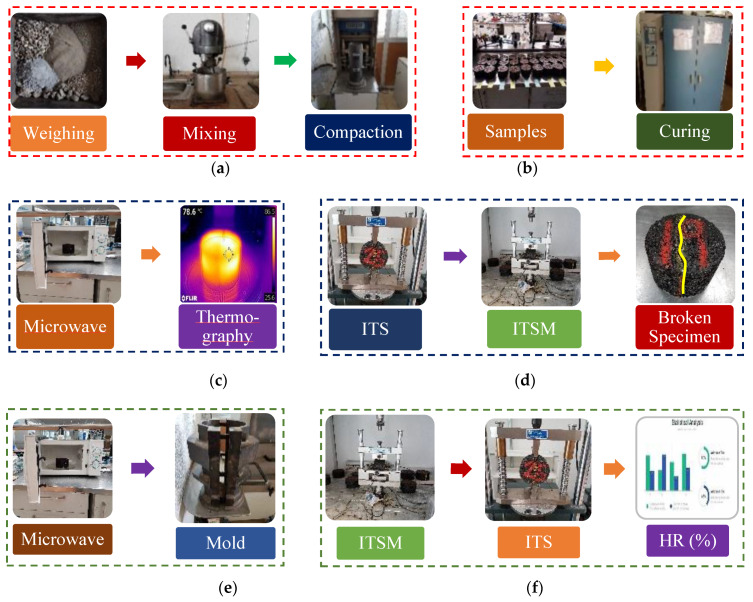
Circular flow chart model of the main phases of the research methodology. (**a**) weighing and dosage of the mixture components (RAP and EAF), and then specimen compaction, (**b**) optimization curing process, (**c**) temperature study, (**d**) initial mechanical tests (indirect tensile strength (ITS) and stiffness modulus), (**e**) novel assisted healing treatment, and (**f**) the repetition of the mechanical tests to calculate the healing ratios using an Excel spreadsheet.

**Figure 2 materials-13-02502-f002:**
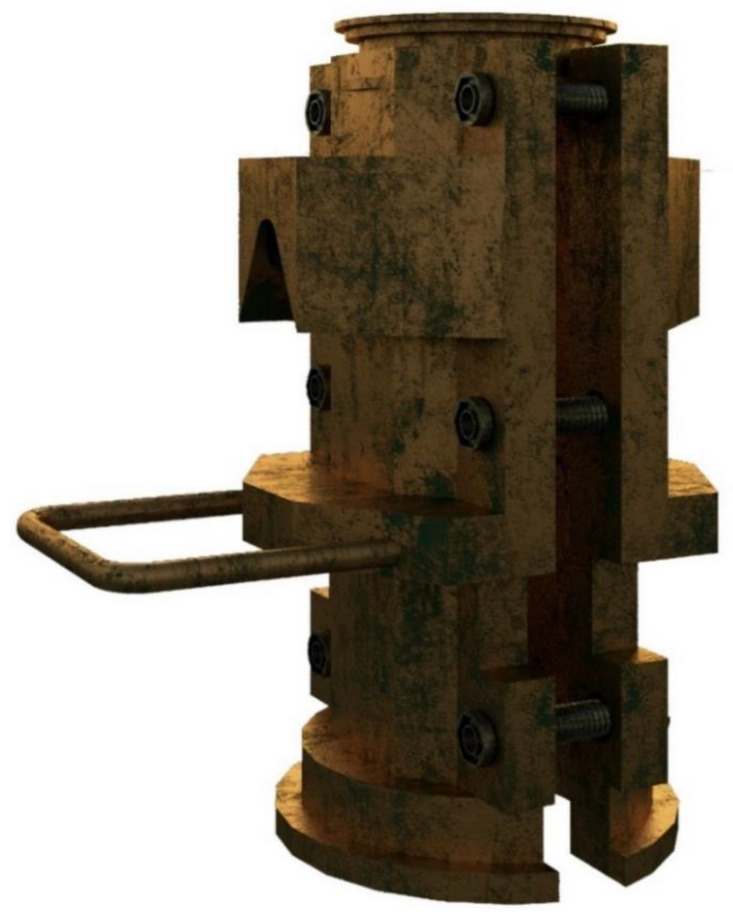
A 3D rendering model of the novel prototype-mold design.

**Figure 3 materials-13-02502-f003:**
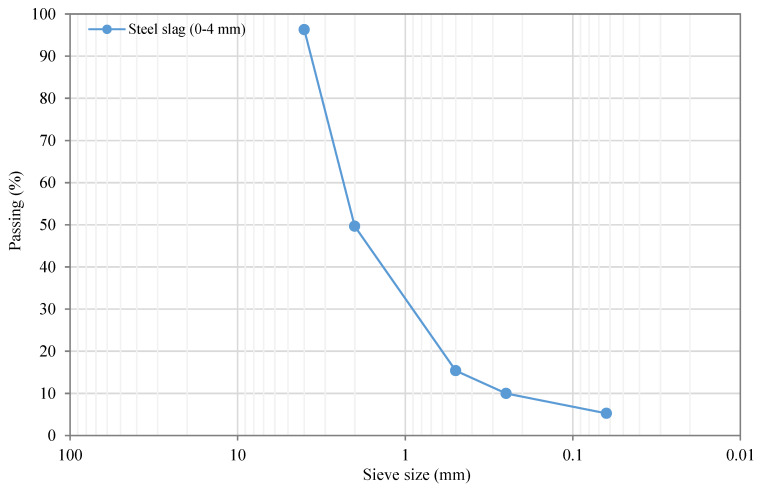
Aggregate grading curve of steel slag passing through the # 4 mm sieve.

**Figure 4 materials-13-02502-f004:**
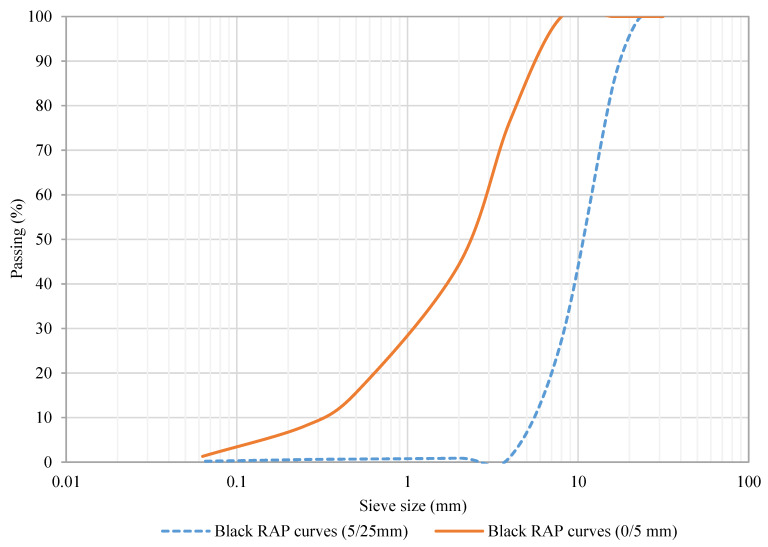
Black grading curves for both recycled asphalt pavement (RAP) fractions.

**Figure 5 materials-13-02502-f005:**
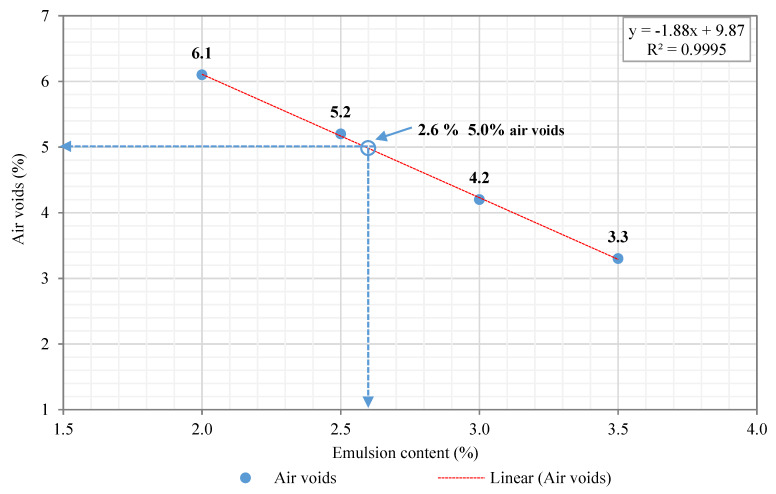
Determination of the optimum emulsion content (OEC) for 5.0% air voids content.

**Figure 6 materials-13-02502-f006:**
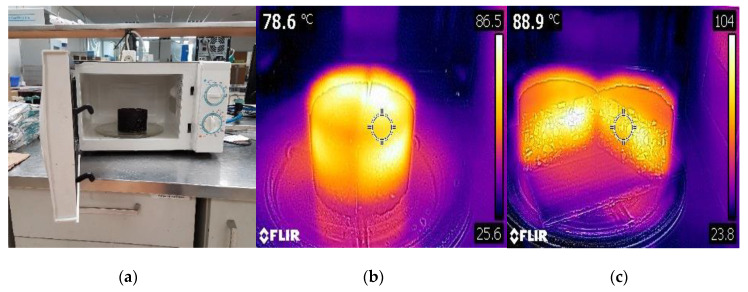
Thermographic analysis: (**a**) Microwave heating oven, (**b**) Surface temperature of the test specimen, (**c**) and internal heating temperature of the specimen.

**Figure 7 materials-13-02502-f007:**
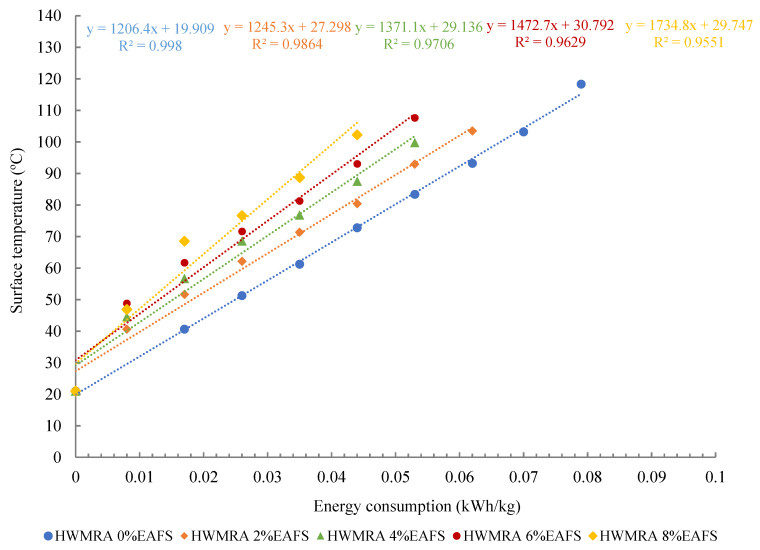
Microwave heating temperature of the specimen’s surface (°C) vs. microwave heating energy consumption (kWh/kg).

**Figure 8 materials-13-02502-f008:**
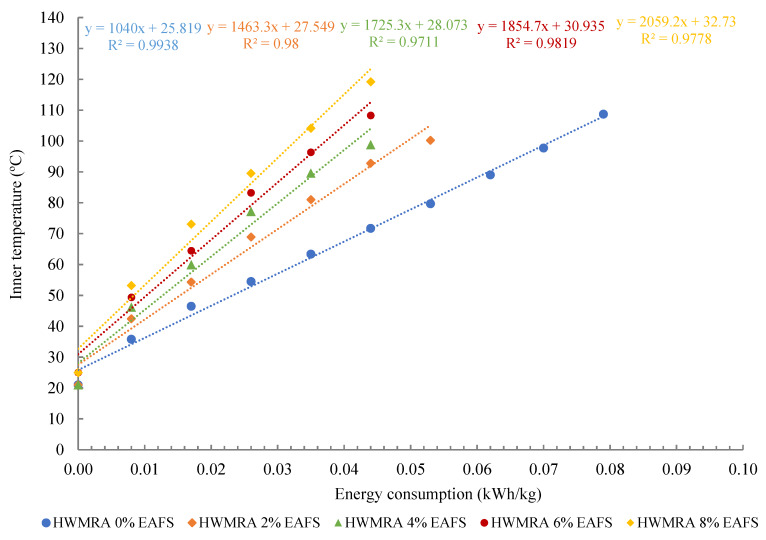
Microwave heating temperature of the inner section of the test specimens (°C) vs. microwave heating energy consumption (kWh/kg).

**Figure 9 materials-13-02502-f009:**
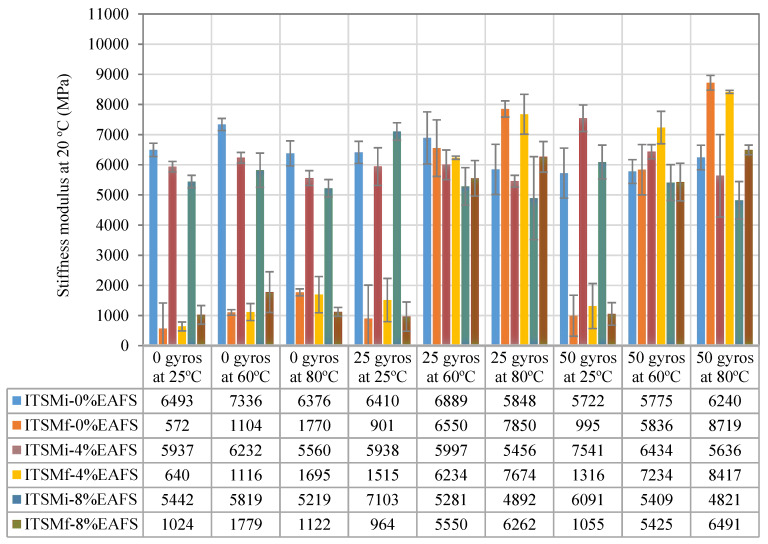
Initial and repeated stiffness modulus value results, at 20 °C, of the EAF slag mixtures.

**Figure 10 materials-13-02502-f010:**
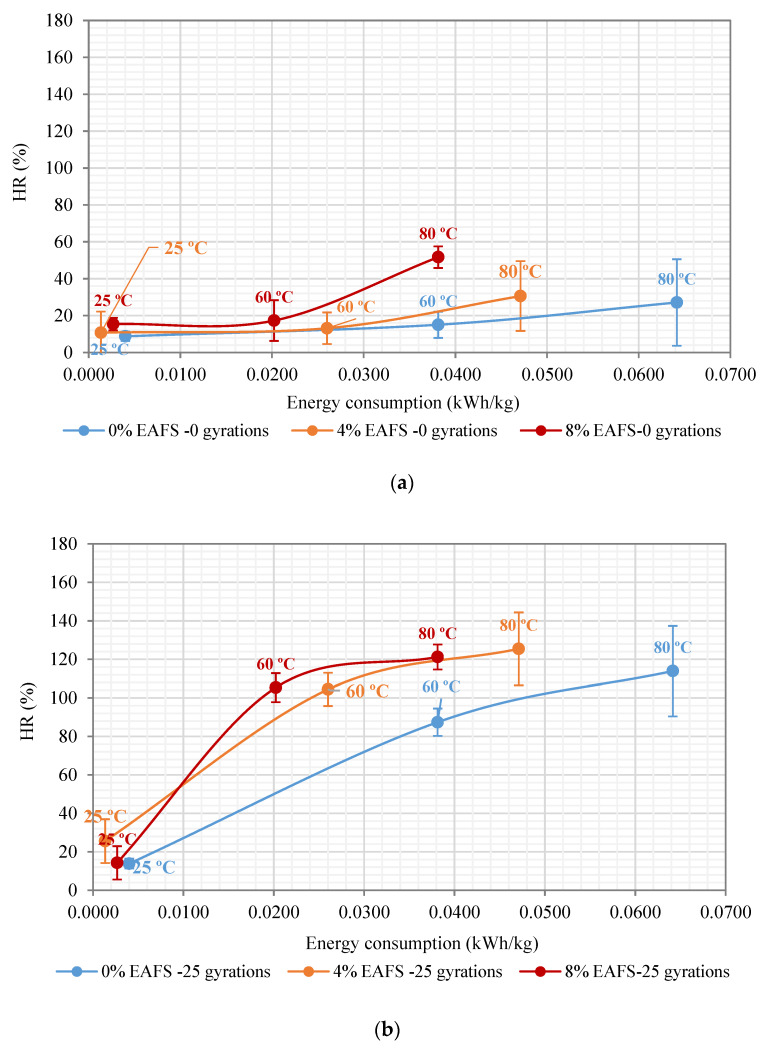
Healing rates (HR) of the stiffness modulus against microwave heating energy consumed (kWh/kg) from each EAF slag mixture content (0%, 4%, and 8%) using three recompaction energy levels: (**a**) 0 gyrations; (**b**) 25 gyrations; and (**c**) 50 gyrations.

**Figure 11 materials-13-02502-f011:**
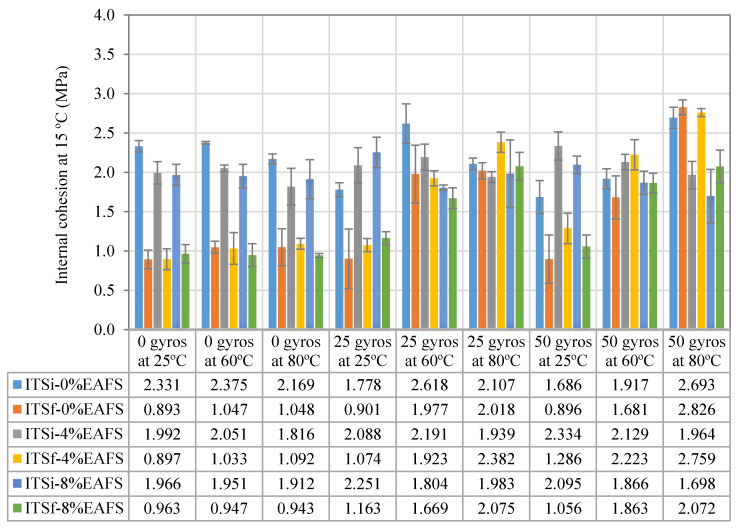
Indirect tensile strength value results, at 15 °C, of the EAF slag mixtures.

**Figure 12 materials-13-02502-f012:**
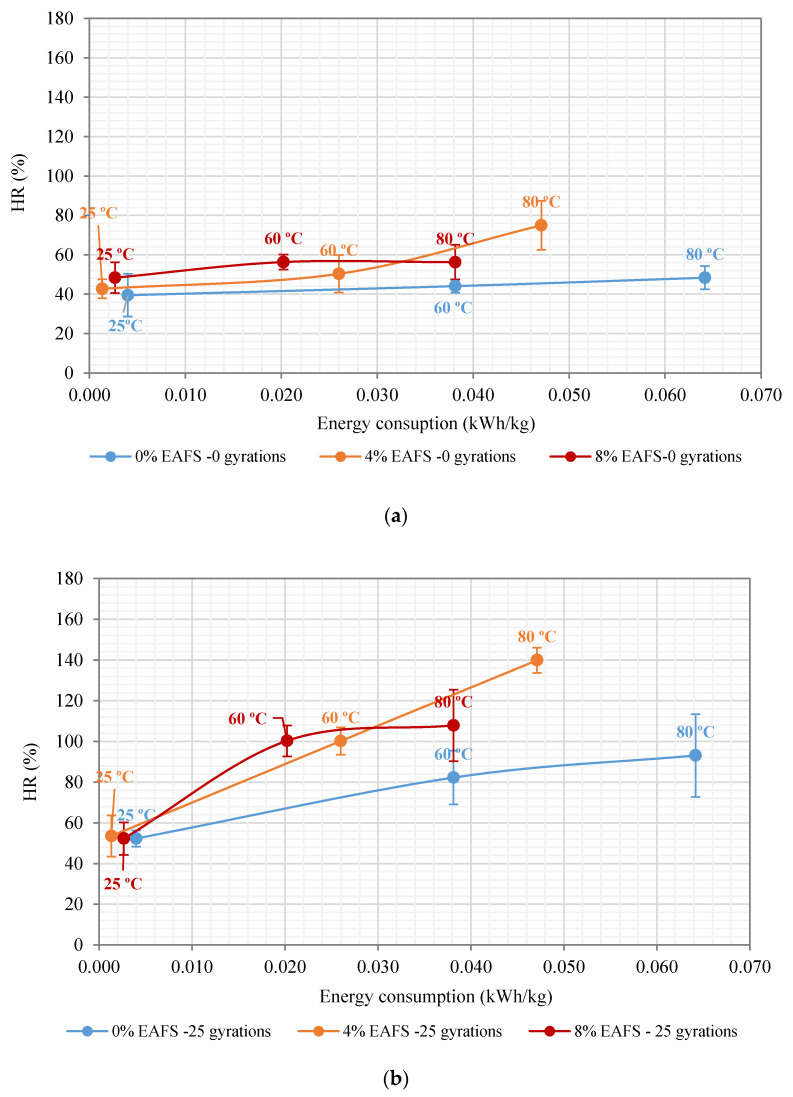
Healing performance rates of indirect tensile strength against microwave heating energy consumption (kWh/kg) for each EAF slag mixture content (0%, 4% and 8%) applying three recompaction energy levels: (**a**) 0 gyrations; (**b**) 25 gyrations; and (**c**) 50 gyrations.

**Table 1 materials-13-02502-t001:** Experimental setup and test matrix: Gyrations vs. microwave temperature and slag content.

Gyrations (Ni)	EAFS (%)
0%	4%	8%
T_25 °C_	T_60 °C_	T_80 °C_	T_25 °C_	T_60 °C_	T_80 °C_	T_25 °C_	T_60 °C_	T_80 °C_
0	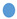	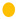	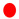	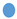	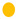	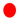	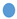	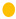	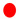
25	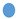	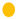	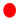	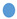	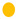	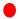	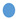	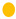	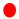
50	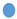	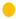	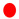	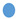	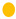	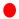	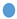	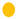	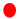

**Table 2 materials-13-02502-t002:** Particle aggregate size distribution of slag aggregates.

Sieve Size (mm)	8	4	2	0.5	0.25	0.063
Passing (%)	100	96.29	49.64	15.40	9.97	5.26

**Table 3 materials-13-02502-t003:** Chemical composition of electric arc furnace (EAF) slag.

Chemical Composition	Values (%)
Al_2_O_3_	8.81
Cao	24.28
Fe_2_O_3_	40.49
MgO	3.02
MnO	4.72
SiO_2_	12.60
P_2_O_5_	0.36
Other substances	5.72

**Key:** EAF slag composition before the hydration process applied and used as an aggregate.

**Table 4 materials-13-02502-t004:** Maximum density and physical characteristics of the recycled RAP binder.

Properties	Unit	Test Method	Value
Maximum density	g/cm^3^	EN 12697-6:2012	2.443
Bitumen content	%, o/RAP	EN 12697-1:2012	4.89
Penetration test	0.1 dmm	EN 1426:2015	11
Softening point	°C	EN 1427:2015	80.3

**Table 5 materials-13-02502-t005:** Volumetric characteristics of the preliminary mixture design.

Mixture Properties	Test Method	Emulsion Content (%, o/RAP)			
		2.0%	2.5%	3.0%	3.5%
Maximum density, (g/cm^3^)	EN 12697-5:2012	2419	2418	2414	2395
Apparent density, by SSD, (g/cm^3^)	EN 12697-6:2012	2269	2292	2313	2339
Geometric density (g/cm^3^)	EN 12697-6:2012	2231	2258	2277	2290
Air voids, Vm, (%)	EN 12697-8:2003	6.1	5.2	4.2	3.3

**Table 6 materials-13-02502-t006:** Composition of the HWRA mixtures with RAP and EAF slag aggregates.

Sieve Size (mm)		100% RAP 0% EAFS		98% RAP + 2% EAFS		96% RAP + 4% EAFS		94% RAP + 6% EAFS		92% RAP + 8% EAFS	
		Mass		Mass (g)		Mass (g)		Mass (g)		Mass (g)	
		(%)	(g)	RAP	EAFS	RAP	EAFS	RAP	EAFS	RAP	EAFS
20	12.5	9.3	93	93	–	93	–	93	–	93	–
12.8	8	9.8	98	98	–	98	–	98	–	98	–
8	4	30.3	303	303	–	303	–	303	–	303	–
4	2	13.8	138	123	22.5	113	37.5	138	60	88	75
2	0.5	16.2	162	159.8	3.3	155	9.9	162	13.2	148.8	19.8
0.5	0	20.6	206	203.2	4.2	198	12.6	206	16.8	189.2	25.2
Weight (g)		100	1000	980	30	960	60	940	90	920	120
Emulsion (%/o, aggregates)		–	2.60	2.65	–	2.69	–	2.74	–	2.78	–
Total weight (g)		–	1026	1037		1047		1057		1068	

**Table 7 materials-13-02502-t007:** Healing rates (HR), microwave heating energy (kWh/kg), and costs of healing efficiency (kWh/kg) for each EAF slag mixture content.

Healing Rates (HR)	EAFS Content (%)	25 Gyrations				50 Gyrations			
		Energy	ΔEc *		Price	Energy	ΔEc		Price
		(kWh/kg)	(kWh/kg)	ΔEs ** (%)	(kWh/kg)	(kWh/kg)	(kWh/kg)	ΔEs (%)	(kWh/kg)
HR = 75% (Stiffmes)	0%	0.0314	0.0178	56.69	0.003762	0.0277	0.0136	49.10	0.003318
	4%	0.0158	0.0022	13.92	0.001893	0.0162	0.0021	12.96	0.001941
	8%	0.0136	–	–	0.001629	0.0141	–	–	0.001689
HR = 100% (Stiffmes)	0%	0.0479	0.0294	61.38	0.005738	0.0407	0.0204	50.12	0.004876
	4%	0.0241	0.0056	23.24	0.002887	0.0242	0.0039	16.12	0.002899
	8%	0.0185	–	–	0.002216	0.0203	–	–	0.002432
HR= 75% (ITS)	0%	0.0278	0.0178	64.03	0.003330	0.0182	0.0097	53.30	0.002180
	4%	0.0121	0.0021	17.36	0.001450	0.0133	0.0048	36.09	0.001593
	8%	0.0100	–	–	0.001298	0.0085	–	–	0.001018
HR=100% (ITS)	0%	0.0642	0.0442	68.85	0.007691	0.0342	0.0142	41.52	0.004097
	4%	0.0261	0.0061	23.37	0.003127	0.0261	0.0061	23.37	0.003127
	8%	0.0200	–	–	0.002396	0.0200	–	–	0.002396

**Key***: Ec = Microwave heating energy consumption (kWh/kg); and ****** Es = represents the microwave heating energy savings, or the amount of energy reduction, expressed in terms of percentage (%).

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
