# Peer review of "Self-Healing Analysis of Half-Warm Asphalt Mixes Containing Electric Arc Furnace (EAF) Slag and Reclaimed Asphalt Pavement (RAP) Using a Novel Thermomechanical Healing Treatment"

_materials, 2020, doi:10.3390/ma13112502_

Round 1

Reviewer 1 Report

In recent years, the self-healing of asphalt pavement is becoming a hot research area. However, most of the studies are only conceptual, and there are still many problems remained in technology, economy and application effect. This paper quantify the healing performance rates (HR) of half-warm recycled asphalt (HWRA) mixtures containing Electric Arc-Furnace (EAF) slag and total recycled asphalt pavement (RAP) rates. To this end, a novel assisted thermomechanical healing treatment (a recompaction-based technique and a microwave-based heating method) was put forward to promote the effect of this treatment on the asphalt mixture’s mechanical performance. Compared with the existing research, this study has better practicability and application prospects. One question is, if a damaged asphalt pavement is reheated in-situ and recompacted, should it be called in situ hot recycling or self-healing?

Author Response

Dear Reviewer,

We are pretty grateful for the useful feedback and comments that were given to us by this Reviewer that, in such a way, helped us to enhance the writing quality of the resubmitted research manuscript. For this reason, we acknowledge the time and the effort that this Reviewer has put into the latest resubmitted edition. In this line, we have carefully answered point-by-point all the comments and questions that have been established by this Reviewer and modified the original manuscript accordingly.

Reviewer 1:

In recent years, the self-healing of asphalt pavement is becoming a hot research area. However, most of the studies are only conceptual, and there are still many problems remained in technology, economy, and application effect. This paper quantify the healing performance rates (HR) of half-warm recycled asphalt (HWRA) mixtures containing Electric Arc-Furnace (EAF) slag and total recycled asphalt pavement (RAP) rates. To this end, a novel assisted thermomechanical healing treatment (a recompaction-based technique and a microwave-based heating method) was put forward to promote the potential effect of this treatment on the mixture's mechanical performance. Compared with the existing research, this study has better practicability and application prospects. One question is, if a damaged asphalt pavement is reheated in-situ and recompacted, should it be called in situ hot recycling or self-healing?

Answer 1:

First of all, we appreciate the positive assessment of our research work, and we want to thank you for the question you have put forth.

As everybody knows, both techniques are well differentiated from each other. For example, the hot in-place recycling (HIR) process mainly consists of applying a specific pre-heating on the damaged asphalt surface layer to achieve the proper softening and then the scarifying (milling) process of the cracked asphalt surface layer of the existing highway pavement. Next, the remixing process is conducted in the twin-shaft asphalt mixer of the machinery. Then, all of the materials (i.e., the recovered pavement, rejuvenator agent, and virgin aggregates - if needed) are inserted in the asphalt mixer. Subsequently, the repaving process (i.e., spreading and compacting) of the rejuvenated asphalt mixture is conducted and then put it back into service — Now, going back to the question set out by the Reviewer — if the asphalt pavement is reheated in-situ and recompaction, should it be called in situ hot recycling or self-healing? The answer to your question is highly dependent on the technique you wish to adopt as a mechanism to repair and rehabilitate the damaged asphalt surface layer. Therefore, and based on the two concepts you described above, we can understand this description you’ve made as assisted "Self-healing" treatment. I hope I’ve answered your question.

Kind Regards

The Authors

Reviewer 2 Report

The paper analyses Half-warm recycled asphalt (HWRA) mixtures containing RAP and EAF slag in which A novel thermomechanical healing treatment was put forward to promote the healing process.

Authors presented an interesting study (particularly interesting the  deep literature review on the self healing issue.), and in general, this paper is written well, the manuscript is not acceptable for publication in the present form. Prior to publication authors should revise their manuscript according to the reviewer comments given in the following lines:

Some comments.

Line 7. The authors affiliation should be in English

Line 195. The role of the AC16 D mixture in the methodology followed by the authors is unclear in this section. Please, clarify.

Lines 186-237. Different stages of the study methods are described, however, authors mention phase (i.e. first phase) and stages (i.e. second stage). The use of a single denomination would improve the clarity of the methods section.

Line 246. The first stage must be (a), as it is referred in Figure 1.

Line 258. Check the font size.

Lines 410-411. Acording to the authors, the inclusion of EAF slag in the asphalt mixture design does not entail any additional cost since the acquisition and production cost of steel slag is similar to that of the sale price of virgin aggregates. However, Does not the EAF replace the RAP material instead virgin aggregates?

Line 571, Figure 6. Were the specimens cut in two halves before or after the have been heated in the microwave oven?

Line 621. Why these values (6465 MPa) do not coincide with those shown in Figure 9.

Line 671: "If" (red colour)

Line 682: As general trend shown by Figure 10, one can...?

Line 855. Revise the spelling

Line 1042. Revise the references. Which is the journal where this work was published?

Author Response

Dear Reviewer,

We are pretty grateful for the useful feedback and comments that were given to us by this Reviewer that, in such a way, helped us to enhance the writing quality of the resubmitted research manuscript. For this reason, we acknowledge the time and the effort that this Reviewer has put into the latest resubmitted edition. For this reason, we have carefully answered point-by-point all the comments and questions that have been established by this Reviewer and modified the original manuscript accordingly.

Reviewer 2: Some comments.

Line 7. The author's affiliation should be in English

Answer:

We appreciate the suggestion. However, according to the latest organic law modification released by the Ministry of Science, Technology, and Innovation, as well as by the Technical University of Madrid, the institutional affiliation (organization and address) of the authors should be placed and appeared in scientific publications in Spanish.

Line 195. The role of the AC16 D mixture in the methodology followed by the authors is unclear in this section. Please, clarify.

Answer:

We thank the Reviewer for the insight. The threshold values selected for the designed asphalt concrete mixture were proposed to fall in the air voids range of 4 and 6%.

Lines 186-237. Different stages of the study methods are described, however, authors mention phase (i.e. first phase) and stages (i.e. second stage). The use of a single denomination would improve the clarity of the methods section.

Answer:

We thank you for bringing this point to our attention. We have used and chosen the word “Phase,” and it was incorporated in Lines, 188, 193, 212, 229, 239, respectively.

Line 246. The first stage must be (a), as it is referred in Figure 1.

Answer:

Thank you for the observation. Therefore, the first phase consisted of weighing and dosing the corresponding proportions of the mixture components (i.e., steel slag, recycled aggregates, and emulsion) that were used in the production of such asphalt mixtures, in which all of the materials were mixed and then compacted with the gyratory compactor. (Page 5, Line 191-195).

Line 258. Check the font size

Answer:

Thank you for the observation. The font size has been changed.

Lines 410-411. Acording to the authors, the inclusion of EAF slag in the asphalt mixture design does not entail any additional cost since the acquisition and production cost of steel slag is similar to that of the sale price of virgin aggregates. However, Does not the EAF replace the RAP material instead virgin aggregates?

Answer:

Thank you for the feedback. So, we have decided to remove this statement from the “Materials” section.

Line 571, Figure 6. Were the specimens cut in two halves before or after the have been heated in the microwave oven?

Answer:

We thank the Reviewer for the comment. Figure 6 illustrates how the asphalt specimens are initially cut into two halves and then placed inside the microwave cavity to conduct the microwave temperature study. Page 16. Lines 568-570

Line 621. Why these values (6465 MPa) do not coincide with those shown in Figure 9.

Answer:

We thank the Reviewer for bringing this point to our attention. The average, standard deviation (SD), and coefficient of variation (CV) values of the stiffness modulus values of the mixtures were recalculated and updated in the resubmitted manuscript edition.

Line 671: "If" (red colour) ---

Answer

Thank you for the observation. The font color has been changed to black color.

Line 682: As a general trend shown by Figure 10, one can...?

Answer:

Thank you for the suggestion. However, it is not entirely clear to me. Would you mind providing further clarification on this phrase?

Line 855. Revise the spelling

Answer:

Thank you for the observation. The spelling has been rewritten to explain the following, “so it would be interesting to determine the optimal temperature at which the healing curve peak begins to stabilize and then fall again.” Page 27. Lines 856-858.

Line 1042. Revise the references. Which is the journal where this work was published?

Answer:

This research work Optimization of Steel Slag Aggregates for Bituminous Mixes in Saudi Arabia has been published in the “Journal of Materials in Civil Engineering.” Where the authors claimed that the pure steel slag aggregate (SSGA) mix showed poor performance in fatigue. The test could not be performed on pure slag mixes at the 689.5 kPa (100 psi) level due to the low tensile strength of slag mixes at high temperatures.

Additionally, according to the research article published by Marta Skaf et al., whose title is “EAF slag in asphalt mixes: A brief review of its possible reuse – Review.” 2.2. Uses. The authors claimed that the behavior of mixtures manufactured using exclusively steel slag was defined as inappropriate (Bagampadde et al., 1999). The mixtures in which the EAF slag was incorporated as filler presented worse performance after submegence, an intolerable fatigue behavior and poorer results in other fields. Furthermore, adhesion with bitumen was visibly better in samples with limestone as fines and filler (Bagampadde et al.,1999). Other authors have also cited this research manuscript entitled “Reuse of steel slag in bituminous paving mixtures.”

We are pretty grateful for the useful feedback and comments that were given to us by this Reviewer that, in such a way, helped us to enhance the writing quality of the resubmitted research manuscript. I hope I’ve responded to each of your comments and questions satisfactorily.

Kind Regards

The Authors

Reviewer 3 Report

First of all, in the opinion of this referee this research paper is appropriate and focused well on the Materials Journal's scope. In this way, only some minor changes must be made, in order to improve the paper.

On the one hand, after reading this article, I believe the article presents a very extensive research and well structured. The subject addressed in this article is worthy of investigation. However, the studied item (Healing Capacity of Asphalt Mixture) is not new.

Consequently, some minor changes should be included. In section 1, entitled "introduction", recently references must be added at the end, because there are very closed with the present research paper, specifically:

https://doi.org/10.1016/j.conbuildmat.2020.119268

https://doi.org/10.3390/ma11050800

Figure 1 must be improved.

In relation to the designed test equipment, researchers must explain how recompaction of specimens can be perform in a real site.

In relation with the loading end condition, the authors must declare the proportion of the maximum load that ends the test.

A graphical abstract should be added to improve the understanding of the research process.

Author Response

Dear Reviewer,

We are pretty grateful for the useful feedback and comments that were given to us by this Reviewer that, in such a way, helped us to enhance the writing quality of the resubmitted research manuscript. For this reason, we acknowledge the time and the effort that this Reviewer has put into the latest resubmitted manuscript edition. In this line, we have carefully answered point-by-point all the comments and questions that have been established by this Reviewer and modified the original manuscript accordingly.

Reviewer 3:

First of all, in the opinion of this referee this research paper is appropriate and focused well on the Materials Journal's scope. In this way, only some minor changes must be made, in order to improve the paper.

On the one hand, after reading this article, I believe the article presents a very extensive research and well structured. The subject addressed in this article is worthy of investigation. However, the studied item (Healing Capacity of Asphalt Mixture) is not new.

Consequently, some minor changes should be included. In section 1, entitled "introduction," recently references must be added at the end, because there are very closed with the present research paper, specifically:

https://doi.org/10.1016/j.conbuildmat.2020.119268

https://doi.org/10.3390/ma11050800

Answer

We appreciate the positive assessment of our work and for bringing all these points to our attention. Both references you suggested have been added to the latest revised manuscript edition. To be more specific, the references were incorporated in the Introduction section as follows:

  1. “Self-healing capacity of asphalt mixtures including by-products both as aggregates and heating inductors.” doi:10.3390/ma11050800

Magnetic induction heating technique that incorporates conductive materials (e.g., slag, steel wool, steel fibers, metal particles, carbon black, and graphite) to speed up the asphalt healing process [18–23]

  1. “Effect of moisture and freeze-thaw damage on microwave healing of asphalt mixes.” https://doi.org/10.1016/j.conbuildmat.2020.119268

Likewise, Kavussi et al., [67] compared the effect of different parameters on microwave healing efficiency of asphalt mixtures using the semicircular bending beam (SCB) and indirect tensile test (IDT). They found that the recovery of mechanical damage of activated carbon modified asphalt mixes reached a healing index peak of 63% (SCB) and 58% (IDT) at 25 ºC, and that the healing index had notably decreased as the fracture-healing cycles increased. (Page 3. Lines 135-139).

  1. Figure 1 must be improved.

Answer: Thank you for the observation. Figure 1 has been improved.

  1. In relation to the designed test equipment, researchers must explain how recompaction of specimens can be perform in a real site. -------

Answer: Thank you for the comment. As you know, the designed test equipment presented in this investigation aimed at investigating whether the mechanical performance properties of the broken specimens can be recovered when employing microwave radiation heating energy and a recompaction-based technique, once the specimens were subjected to mechanical testing (i.e., ITS and ITSM). However, for full-scale applications, the most traditional techniques that are used to repair the damaged asphalt surface layers are either hot in-place recycling or radiation heating energy through the use of a portable microwave power unit. This latter treatment can be combined with the use of conventional machinery (e.g., a vibratory double-drum and a pneumatic tired road roller compactor) for the recompaction of the cracked asphalt surface layer.

In relation with the loading end condition, the authors must declare the proportion of the maximum load that ends the test---------------

Answer:

As for the indirect tensile strength, the recommended constant deformation rate applied to the asphalt specimens was 50 ± 2 mm/min, according to EN 12697-23.

The ITS test was interrupted when the peak compressive strength load dropped by 20% to avoid excessive deformation of the specimens. (Page 10. Lines 389-390). Besides, it is worth mentioning that the maximum load recorded throughout the ITS test might vary depending on the slag content (0%, 4%, and 8%) added to the mixture design, as well as the number of microwave healing cycles to which the specimen is exposed.

  1. A graphical abstract should be added to improve the understanding of the research process.

Answer: Thank you for the suggestion. The graphical abstract has already been uploaded in the submission process.

Thank you for the contributions and the comments that helped us to improve the manuscript quality.

Kind Regards

The Authors

Reviewer 4 Report

The authors have put a lot of effort into testing samples and analyzing data. The test results could be useful if some statements in this paper are clearly explained. Following are the comments that the authors need to clarify:

  1. Why did the authors try to test tensile strength of mixtures? Why not rutting or fatigue?

  1. Can authors consider different heating power than 400W?

  1. Is it expected to observe better healing efficiency if more than 8% EAF slag is used?

  1. The authors mentioned at 80ËšC, the 4% EAf mixture show high cohesion. My question is how such temperature is going to create on asphalt surface in the field? It is not practical in most places.

Author Response

Dear Reviewer

We are pretty grateful for the useful feedback and comments that were given to us by this Reviewer that, in such a way, helped us to enhance the writing quality of the resubmitted manuscript. For this reason, we acknowledge the time and the effort that this Reviewer has put into the latest submitted edition. In this sense, we have carefully answered point-by-point all the comments and questions that have been established by this Reviewer and modified the original manuscript accordingly.

Reviewer 4

The authors have put a lot of effort into testing samples and analyzing data. The test results could be useful if some statements in this paper are clearly explained. Following are the comments that the authors need to clarify: 

  1. Why did the authors try to test the tensile strength of mixtures?

As you know, the lack of cohesion in the mixture design can lead to the loss of aggregate and to peeling phenomenon, both of which can considerably reduce the resistance of the mix to tangential stresses and strains. For this reason, we’ve decided to evaluate this property by means of the indirect tensile strength, together with the stiffness modulus of the asphalt mixture, according to EN 12697-26. What’s more, most of the research studies found in the literature review have tried to quantify these properties using the indirect tensile strength test and stiffness modulus.

  1. Why not rutting or fatigue?

As you know, rutting is not a critical surface pavement distress associated with mixtures containing total RAP contents above 90% of the total weight of the aggregates. Since they possess high resistance to rutting or permanent deformation because of the presence of aged binder, as reported by multiple research studies elsewhere. On the other hand, why not fatigue? It’s a good question. Both indirect tensile fatigue test (ITFT) and four-point bending (4PB) beam test will be investigated for future research works.

  1. Can authors consider different heating power than 400W?

We have not considered working with different heating power levels. However, it would be interesting to work with different heating power levels to assess the healing/recovery ratios of the slag mixtures as well as the healing efficiency spent by the microwave oven during the heating process. In this regard, other researchers have reported similar trends in terms of setting the microwave oven at the medium power level. For instance, Gulisano et al, has worked with a microwave power level of 350 W to heat the samples of the asphalt mixtures. doi:10.3390/app10041428

  1. Is it expected to observe better healing efficiency if more than 8% EAF slag is used?

According to our findings, we’ve found that the stiffness modulus of the slag mixtures began to decrease as the slag mixture content increases. So, it is recommended not to add more than 8% of the total volume of the aggregates in the mixture design.

  1. The authors mentioned at 80ËšC, the 4% EAf mixture show high cohesion. My question is how such temperature is going to create on asphalt surface in the field? It is not practical in most places.

Nowadays, there are several efforts underway to develop a mobile microwave power heating unit that is capable of heating the damaged asphalt surface layer at temperatures of a conventional HMA mixture. According to Bosisio et al., assessed the “Asphalt Road Maintenance with a Mobile Microwave Power Unit”. They reported that thanks to the use of a microwave power unit, the surface heating temperatures were similar to a conventional hot asphalt mixture.

We appreciate the positive assessment of our research work and for the comments and feedback that, in such a way, helped us to improve the quality of the revised manuscript.

Kind regards

The Authors